# Many phases of generalized 3D instanton crystals

**Matti Jarvinen[1,2,3]⋆, Vadim Kaplunovsky[4]† and Jacob Sonnenschein[1]‡**

**1** The Raymond and Beverly Sackler School of Physics and Astronomy,
Tel Aviv University, Ramat Aviv 69978, Tel Aviv, Israel
**2** Asia Pacific Center for Theoretical Physics, Pohang 37673, Rebublic of Korea
**3** Department of Physics, Pohang University of Science and Technology,
Pohang 37673, Republic of Korea
**4** Physics Theory Group and Texas Cosmology Center, University of Texas,
Austin, TX 78712, USA

⋆ matti.jarvinen@apctp.org, † vadim@physics.utexas.edu, ‡ cobi@tauex.tau.ac.il

---

**NOTE:** This document includes interactive 3D versions of several figures, for which you will need a compatible pdf reader.

---

## Abstract

Nuclear matter at large number of colors is necessarily in a solid phase. In particular holographic nuclear matter takes the form of a crystal of instantons of the flavor group. In this article we initiate the analysis of the three-dimensional crystal structures and the orientation patterns for the two-body potential that follows from holographic duality. The outcome of the analysis includes several unexpected results. We perform simulations of ensembles of $\mathcal{O}(10000)$ instantons whereby we identify the lattice structure and orientations for the different values of the weight factors of the non-Abelian orientation terms in the two-body potential. The resulting phase diagram is surprisingly complex, including a variety of ferromagnetic and antiferromagnetic crystals with various global orientation patterns, and various "non-Abelian" crystals where orientations have no preferred direction. The latter include variants of face-centered-cubic, hexagonal, and simple cubic crystals which may have remarkably large or small aspect ratios. The simulation results are augmented by analytic analysis of the long-distance divergences, and numerical computation of the (divergence free) energy differences between the non-Abelian crystals, which allows us to precisely determine the structure of the phase diagram.

# 1 Introduction and Summary

It is believed that real life $N_c = 3$ QCD cold nuclear matter forms a quantum liquid, but for large $N_c$ it becomes a crystalline solid [1]. This structure of nuclear matter has been intensively investigated in the context of Skyrme theory [2–4]. A more modern approach to nuclear matter at large $N_c$ is based on the use of holography. In holography mesons are open string configurations with endpoints on flavor branes. Baryons have the structure of a baryonic vertex connected with $N_c$ strings to flavor branes. It was shown in [5] that the baryonic vertex can be realized in terms of a wrapped $D_p$ brane over a p-cycle with a RR flux of $N_c$. Such structures have been analyzed in a confining holographic background in [6]. A prototype model that admits this structure is the so called Witten-Sakai-Sugimoto model (WSS) [7,8]. The holographic dual of the pure glue theory is based on near horizon limit of $D_4$ branes compactified along one space dimension with anti-periodic boundary conditions for the fermions. The quark sector is mapped into stacks of $D_8$ and $\bar{D}_8$ probe flavor branes that are asymptotically separated and merge in the IR region. This model was then generalized to include non-antipodal flavor branes [9]. It was shown in [10], using energy considerations, that in that model the baryonic vertex is immersed in the flavor branes. In such a case the baryon can be viewed as a $U(N_f)$ instanton [11] in a curved Euclidean 4D built from the three ordinary space dimensions and the holographic dimension. At leading order in the strong coupling limit these instantons can be approximated by the BPST instantons. Thus holographic nuclear matter takes the form of a *lattice of BPST instantons*. A key ingredient for the structure of the latter is the interaction between the instantons. It was shown in [1] that the interaction in the WSS model is repulsive at any separation distance. For long distances between the baryons the repulsion is due to the fact that the lightest isoscalar vector, whose exchange yields repulsion, is lighter than the lightest scalar that yields attraction. In the near and intermediate zones the interaction between two instantons of the WSS model is purely repulsive. By using the generalized WSS model instead of the original model, the severeness of the problem can be reduced. As was shown in [1], only in the generalized model (and not in the original WSS) there is, in addition to the repulsive force, also an attractive one due to an interaction of the instantons with a scalar field that associates with the fluctuation of the embedding, though the ratio of the attractive to repulsive potential can never exceed 1/9. In [12] it was shown that there is another holographic model [13] with a dominance of the attraction at long distances, together with a small binding energy, as one encounters in nature.

Thus, the crystalline of holographic nucleons has to be a multi-instanton solution of the ADHM equations. Motivated by the interesting behavior of skyrmions at high density and being equipped with the new methodology of holography, first steps in determining the structure of the cold nuclear matter were made in a context of a 1D toy model in [14] and [15]. For the straight 1D lattices, it was found that the orientations are periodically running over elements of a $\mathcal{Z}_2$, Klein, prismatic, or dihedral subgroup of the $\frac{SU(2)}{\mathcal{Z}_2}$, as well as irrational but link-periodic patterns. In addition we discovered a formation of zigzag-shaped lattices, where instantons are popping up to the holographic direction. For these configurations we detected 4 distinct orientation phases — the anti-ferromagnet, another Abelian phase, and two non-Abelian phases. We further discussed the phase diagram of 2D instantons [16].

Related studies appeared in [17–44]. In particular, [18] and [19] have analyzed the phase diagram of configurations with finite baryon density, treating the instantons as point-like sources. A completely different approach was taken in [19,40–42], where the instanton structure was replaced by a homogeneous non-Abelian field, motivated by dense instanton configurations smeared over spatial directions.

In analogue to the 1D toy model, realistic holographic models of nuclear matter are expected to involve 3D and 4D crystals and a transition between them with increasing density. In this article, we will only discuss the 3D crystals. But solving even the 3D ADHM equations is a tremendous task. Instead in this paper we use an approximation of a two body instanton interaction which is adequate for crystals of low density. In [15] and in [24] the two body potential between two instantons was derived. We write down a more general potential depending on two parameters $\alpha$ and $\beta$,

$$
\begin{aligned}
\mathcal{E}^{2\,\text{body}}(m,n;\alpha,\beta) = {} & \frac{2N_c}{(1+2\alpha+2\beta)\lambda M}\frac{1}{|X_n - X_m|^2} \\
& \times\left[\frac{1}{2} + \alpha\,\text{tr}^2\left(y_m^\dagger y_n\right) + \beta\,\text{tr}^2\left(y_m^\dagger y_n(-i\vec{N}_{mn}\cdot\vec{\tau})\right)\right],
\end{aligned} \tag{1}
$$

such that the potential discussed in the earlier reference is obtained for $\alpha = \beta = 1$. Here in this research work we generalize the potential by letting $\alpha$ and $\beta$ vary. The notation in this potential will be explained in detail below. At the moment it is enough to know that the $SU(2)$ matrices $y_m$ and $y_n$ carry the information on the orientations of the instantons $n$ and $m$. In this way we consider different weights for the orientation terms in the potential.

In this paper we determine the crystal structure and the instanton orientation patterns that minimize the total energy of the system. In order to understand our results, it is important to notice that the two-body interaction (1) gives rise to a strong long-distance (IR) divergence in three dimensions. The potential due to a configuration of (large) size $R$ behaves as $\int dr\, r^2\,\mathcal{E}^{2\,\text{body}}(r) \sim \int dr\; r^0 \sim R$ so it is linearly divergent. Therefore long distance effects are strongly enhanced, leading to drastically different results in comparison to the results for lower dimensions [15, 16]. In particular, much of the phase structure is determined through the comparison of the coefficients of the divergent term for various orientation structures. In several of these phases, minimization of the energy leads to nontrivial long distance correlations between the instanton orientations: for example, we obtain orientation structures which are spherical at long distances.

Because of the divergence we need a long-distance cutoff in all our simulations and computations, so we need to be particularly careful that our results are independent of the cutoff. Moreover, in case of simulations, the long distance interactions lead to clustering of the instantons at the surface of the simulation volume if we only set a hard wall cutoff which forces the instantons to stay within the volume. The removal of this undesired effect necessitates the use of a smooth external force which pushes the instantons towards the center of the simulation. Because of the long-distance divergence, however, there is no obvious "correct" choice for the external force for all configurations that we consider. A simple choice which is applicable to all configurations that we encounter is to take a force which sets the (locally averaged) instanton density to be constant. For most of the phase diagram this force matches with the (regularized) force due to a homogeneous density of instantons outside the simulation volume, i.e., the force due to the instantons left out of the simulation. Notice also that all our simulation results turn out to be insensitive to the precise choice of the force.

We are now ready to summarize our results. The results are obtained by using three different methods of analyzing the lattices of instantons:

(i) We perform simulations of ensembles of instantons of the order of $\mathcal{O}(10000)$ instantons subjected to the two body interactions between any two of them. In addition, in order to reduce boundary effects, we also added an external force which, as discussed above, sets the average density of instantons to be constant. Using these simulations were were able

to determine the geometry and orientation structures of the instanton collection with the lowest total energy.

(ii) We determine the orientation patterns based on the behavior of the two body potential for far apart instantons, i.e., the long-distance divergence discussed above. In this range of distances between the instantons we take the continuum limit and ignore the lattice geometry. We compute the total energy of the system as a function of $\alpha$ and $\beta$ and in particular determine the associated lowest energy configuration.

(iii) We compute the energy difference between various pairs of geometrical and orientation structures by which we eventually determine the phase diagram. The computations of energy differences yield finite results since the divergences are cancelled out in the differences.

The main outcome results of the simulations are the following for the basic cases:

- In the *non-orientation case $\alpha = \beta = 0$* the crystal structure is face-centered-cubic (fcc).

- *The basic oriented case $\alpha = \beta = 1$* is a face-centered tetragonal lattice with a large aspect ratio,[1] i.e., fcc with one direction rescaled, breaking the cubic symmetry. The aspect ratio $c$ is large: we find $c \approx 2.467$. That is, the instantons form clearly *separated layers with two-dimensional square lattice structure*. A sample of this structure is shown in fig. 2. The two dimensional layers have antiferromagnetic structure. The orientations between the layers repeat in cycles of two, as seen from fig. 2. Overall, there are therefore four distinct and linearly independent orientations so that the set of orientations does not single out any direction. We call this class of orientation structures "non-Abelian".

We have also revealed the structure of the phase diagram of the ensembles of instantons as a function of $\alpha$ and $\beta$ in (1). The results are mostly based on simulations, with the boundaries between the phases refined by using the methods (ii) and (iii). We identify the following phases:

- *Oriented non-Abelian phase*. This phase contains crystals in the same class with the case $\alpha = 1 = \beta$, i.e., the orientations span the whole four dimensional orientation space. We find this phase in the region given by the conditions $\alpha > \beta$, $\beta > -1/8$, and $\alpha > \gamma_1 \beta$ with $\gamma_1 \approx 0.0575$. The phase is further divided into subphases having different lattice and orientation structures, which can be classified in the following classes:

  (a) Tetragonal/cubic (fcc or fcc-related) lattices with "standard" orientation pattern

  (b) Tetragonal/cubic (fcc or fcc-related) lattices with "alternative" orientation pattern

  (c) Hexagonal lattices

  (d) Simple cubic/tetragonal lattices.

  In the classes (a), (b), and (d) the crystals have patterns of four distinct orientations, whereas the hexagonal lattices have six orientations (details will be given in Sec. 5). We find crystals in all of theses classes also with nontrivial aspect ratios, i.e., with the spatial structure rescaled in one coordinate directions. We also identify regions in the phase diagram where the crystal is exactly fcc (aspect ratio is equal to one) within the class (a), and regions where the crystal is exactly simple cubic, class (d). Moreover we find a line where the crystal is exactly body-centered-cubic (bcc) for the class (b) (aspect ratio is

---

[1]The precise definition of the aspect ratio will be given in Sec. 5

$1/\sqrt{2}$ in fcc units). The two basic cases (non-oriented $\alpha = 0 = \beta$ and oriented $\alpha = 1 = \beta$) are included in the non-Abelian phase as special limiting cases.

- *Anti-ferromagnetic phase*. This phase is found in the sector $\alpha < \beta < \Gamma\alpha$ with $\Gamma \approx 2.14$. We find that the orientation structure is locally antiferromagnetic: in domains of small size, only two different orientations appear. Globally the orientations are arranged spherically. The detailed description of the global structure will be given below. The spatial lattice has a strong tendency of forming spherical shells. The two-dimensional lattice structure depends on $\alpha$ and $\beta$. For $\alpha = \mathcal{O}(1)$ and $\beta = \mathcal{O}(1)$, the spherical shells have the structure of two-dimensional square lattice (with antiferromagnetic orientations). As $\alpha$ and $\beta$ decrease, the lattice changes to rhombic. For small values, $\alpha = \mathcal{O}(0.1)$ and $\beta = \mathcal{O}(0.1)$, the layers are closer to triangular and the spatial structure is locally close to bcc.

- *Spherical ferromagnetic phase*. In the sector $\beta > \Gamma\alpha > 0$, we find a pattern which is locally fcc and ferromagnetic: all nearby orientations are (almost) the same. However globally the orientations form a spherical structure. The simulations indicate that the lattice structure is independent of $\alpha$ and $\beta$ within the phase.

- *Global ferromagnetic phase*. In the area of negative $\alpha$, precisely limited by the conditions $-1/8 < \alpha < 0$, $\beta > -1/8$, and $\alpha < \gamma_2\beta$ with $\gamma_2 \approx 0.543$, we find a global ferromagnetic pattern. That is, the lattice is (locally and globally) fcc and all orientations are aligned.

- *Ferromagnetic egg phase*. In the remaining corner of the phase diagram, i.e., when $\gamma_2\beta < \alpha < \gamma_1\beta$ and $\beta > -1/8$, we find a special ferromagnetic crystal. It is again locally fcc, and the interior of the sample orients as a global ferromagnet, but there is crust of finite width where the orientations obey a global spherical alignment.

The methods that we have developed in this paper for the analysis of holographic nuclear matter, namely, the long distance analysis of the potential, determining the stable crystalline and orientation structure by performing simulations and the computations of energy differences, could be applied to other non-Abelian lattices.

The paper is organized as follows: In the next section we spell out the basic setup of the lattices of instantons. In particular we write down the two body interactions between the instantons, their generalization and the symmetries of the system. In section §3 we describe the outcome of the simulations. We start with the unoriented case. Next we present the results of the standard orientation of the unmofified model of $\alpha = \beta$. Oriented non-Abelian crystals with $\alpha \geq \beta$ come next. For the range of $0 < \alpha < \beta$ an oriented (anti) ferromagnetic spherical phase was found and finally global ferromagnetic phases at $\alpha < 0$. Section §4 is devoted to the computation of the potential and total energy at long distances for the different types of orientations. The phase diagram of the non-Abelian crystals is determined in section §5. We conclude and write down several open questions in §6. Two appendices were added. In the first we determine the potential for a ball of homogeneous matter and in appendix B we present numerical details of the setup of the simulations and on the computations of the energy differences.

## 2 The basic three dimensional setup

We consider a three-dimensional system of point-like $SU(2)$ instantons so that all instantons are located at the same value of the holographic coordinate. Therefore each instanton is characterized in terms of the position $\vec{X} = (X^1, X^2, X^3)$ and the unimodular quaternion $(y^1, y^2, y^3, y^4)$

carrying the information on the orientations of the instantons. The basis vectors in the quaternion space are denoted as usual: $1 = (1, 0, 0, 0)$, $i = (0, 1, 0, 0)$, $j = (0, 0, 1, 0)$, and $k = (0, 0, 0, 1)$. We also use the representation of the quaternions as a 2×2 matrices,

$$y = y^4 + i\tau^j y^j \, , \tag{2}$$

where $\tau^j$ are the Pauli matrices. Then unimodularity is equivalent to the condition $\text{tr}(y^\dagger y) = 2\sum_{k=1}^4 (y^k)^2 = 2$. The ensemble is described in terms of the pairs $(\vec{X}_n, y_n)$ with $n = 1, 2, 3, \dots N$.

## 2.1 Two body forces between instantons

In holographic nuclear physics, like in real life, besides the two-body nuclear forces due to meson exchanges, there are significant three-body and probably also higher number-body forces,

$$\hat{H}_{\text{nucleus}} = \sum_{n=1}^A \hat{H}^{1\,\text{body}}(n) + \tfrac{1}{2} \sum_{\substack{\text{different} \\ m,n=1,\dots A}} \hat{H}^{2\,\text{body}}(m,n) + \tfrac{1}{6} \sum_{\substack{\text{different} \\ \ell,m,n=1,\dots A}} \hat{H}^{3\,\text{body}}(\ell,m,n) + \cdots , \tag{3}$$

where $n$ stands for the quantum numbers of the $n^{\text{th}}$ nucleon.

In the low-density regime of baryons separated by distances much larger then their radii, the two-body forces dominate the interactions, while the multi-body forces are smaller by powers of $(radius/distance)^2$. The dominance of the two body forces for low density was proven in [15]. The two body interaction was also determined there by combining the non-Abelian and the Coulomb energies of the system and re-organizing the net energy into one-body, two-body, etc, terms.

We use a two-body potential motivated by the Witten-Sakai-Sugimoto model [8]. Assuming that the separation $|X_m - X_n|$ between the instantons $n$ and $m$ satisfies

$$\frac{1}{M\sqrt{\lambda}} \ll |X_m - X_n| \ll \frac{1}{M} \, , \tag{4}$$

where $\lambda$ is the coupling constant and $M$ is the Kaluza-Klein scale, the two-body potential in this model takes a simple form [15, 24]:

$$\mathcal{E}^{2\,\text{body}}(m,n) = \frac{2N_c}{5\lambda M} \times \frac{1}{|X_m - X_n|^2} \times \left[ \frac{1}{2} + \text{tr}^2\left(y_m^\dagger y_n\right) + \text{tr}^2\left(y_m^\dagger y_n (-i\vec{N}_{mn} \cdot \vec{\tau})\right) \right] , \tag{5}$$

where we defined the unit vector

$$\vec{N}_{mn} = \frac{\vec{X}_n - \vec{X}_m}{|X_n - X_m|} \, . \tag{6}$$

The lower limit in (4) arises from the instanton size and the upper limit from neglecting curvature corrections in the AdS geometry. Notice that the range of validity is parametrically large in the strong coupling limit $\lambda \to \infty$, where the dual description in terms of classical gravity is reliable. The normalization in (5) was chosen such that taking the average over the orientations of either of the instantons (with the obviously chosen measure) gives

$$\langle \mathcal{E}^{2\,\text{body}}(m,n) \rangle = \frac{N_c}{\lambda M} \frac{1}{|X_n - X_m|^2} \, . \tag{7}$$

Note that the expression inside '$[\cdots]$' is always positive, so the two-body forces between the instantons are always repulsive, regardless of the instantons' $SU(2)$ orientations. However, the orientations do affect the strength of the repulsion: two instantons with similar orientations repel each other 9 times stronger then the instantons at the same distance from each other but whose orientations differ by a 180° rotation (in $SO(3)$ terms) around a suitable axis. *This fact will be at the core of our analysis of instanton crystals in subsequent sections.*

The total energy of the system (including only the dominant two-body terms) is the sum over all pairs:

$$\mathcal{E}_{\text{tot}} = \sum_{n<m} \mathcal{E}^{2\,\text{body}}(m,n)\,. \tag{8}$$

## 2.2 Generalizations of the defining instanton interactions

It is natural to consider a generalization of the two-body potential which includes both the oriented interaction (5) and the orientation independent potential of (7). We therefore define

$$\begin{aligned}
\mathcal{E}^{2\,\text{body}}(m,n;\alpha,\beta) &= \frac{2N_c}{(1+2\alpha+2\beta)\lambda M}\frac{1}{|X_n-X_m|^2} \\
&\times \left[\frac{1}{2} + \alpha\,\text{tr}^2\left(y_m^\dagger y_n\right) + \beta\,\text{tr}^2\left(y_m^\dagger y_n(-i\vec{N}_{mn}\cdot\vec{\tau})\right)\right],
\end{aligned} \tag{9}$$

where the normalization was chosen such that after averaging over the orientations, (7) holds. Notice that $\alpha=0=\beta$ gives the unoriented potential and $\alpha=1=\beta$ gives back (5).

Let us then comment on how the parameters $\alpha$ and $\beta$ affect the interaction in physical terms. The $\alpha$ coefficient multiplies a term where the dependence on orientation is independent of the spatial interactions. For positive (negative) $\alpha$ perpendicular (parallel) spins of nearby neighbors are preferred, and the effect increases with increasing $|\alpha|$. The interaction involving $\beta$ is a bit more complicated since there is nontrivial coupling between the orientations and directions in coordinate space. Picking an instanton with unit orientation ($y=1$), positive (negative) $\beta$ means that the orientation $y$ of a neighboring instanton which is perpendicular (parallel) to the spatial link between the two instantons is preferred. (This with the understanding that the orientations are mapped to the directions in the coordinate space in the standard fashion: for example, the orientation $y=i$ is parallel to the $x$ axis in coordinate space.) We will comment on how these observations are reflected in the structure of the final phase diagram in Sec. 5.

## 2.3 Symmetries of the interaction

The two body potential (5) and its generalizations (9) for various values of $\alpha$ and $\beta$ are invariant under several transformations.

- Since the potential dependence on the space coordinates is via $\vec{X}_n-\vec{X}_m$ and $N_{ij}$ it is obvious that it is invariant under space translation of the locations of the instantons;

$$\vec{X}_n \to \vec{X}_n + \vec{A}\,, \tag{10}$$

  for any constant three dimensional vector $\vec{A}$.

- Whereas the non-orientation part and also for the case of $\beta=0$ the potential is invariant under global rotations. Since $\vec{N}_{ij}$ is not invariant under rotations the full potential is invariant only under rotation which is accompanied with a rotation in the $y$ space described below.

- Changing the sign of either $y_m$ or $y_n$ does not have an effect on the energy. That is, the orientations matter only up to a sign.

- The $y_n$s reside on an $S^3$ and hence are invariant under $SU(2)_L \times SU(2)_R$ symmetry transformations related to the orientation structure, one of which is (in general) coupled to the spatial rotations. First, notice that the interactions only depend on the combination $y_m^\dagger y_n$ which we call the *twist*. Therefore, obviously a left multiplication of all quaternions $y_n$ by a unimodular quaternion $u$ leaves the potential invariant:

$$y_m \to u\, y_m \qquad u \in SU_L(2) \qquad y_m^\dagger y_n \to y_m^\dagger y_n \,. \tag{11}$$

- The other $SU_R(2)$ rotation is coupled to spatial rotations as follows. Under rotation by an angle $\theta$ around the axis $\vec{n}$, the orientation dependent factor in (9) transforms as

$$\vec{N}_{mn} \cdot \vec{\tau} \;\mapsto\; \exp(i\theta\vec{n} \cdot \vec{\tau}/2)\,(\vec{N}_{mn} \cdot \vec{\tau})\,\exp(-i\theta\vec{n} \cdot \vec{\tau}/2) \,. \tag{12}$$

  We therefore identify the action of the rotations to the quaternions as the following right multiplication:

$$y \mapsto y\, \exp(-i\theta\vec{n} \cdot \vec{\tau}/2) \,, \tag{13}$$

  which, when applied together with the spatial rotation, leaves the energy invariant.

- In the special case $\beta = 0$, spatial rotations are decoupled from the orientations; the vectorial transformation (13) and spatial rotations are good symmetries separately.

## 3 Simulations of the various instanton lattices

We have studied the three dimensional crystal structures arising from the above interactions by using numerical simulations. Starting from a random initial condition with $N = \mathcal{O}(10000)$ of instantons in a spherical cavity, we use an algorithm to relax the system towards the state of lowest energy. In more detail, we use a setup which roughly corresponds to adding masses and drag forces for the instantons, and simulating their dynamics as the system converges to a final state with low energy. We also add a compensating "external" force, which (as we explained in the introduction) prevents the instantons from clustering at the edge of our simulation volume. To be precise, we choose a force which sets the density of instantons, averaged locally over small distances, to be constant (see Appendix A). This means that we compute the force field generated by the sample in the continuum limit, i.e., by approximating the instanton configuration by homogeneous matter of constant density, and apply the force of the same magnitude but opposite direction on the instantons during the simulations. We note that for many of the crystals which we find (including the global ferromagnetic and non-Abelian crystals defined below) the force that sets the instanton density to constant is apparently the unique natural choice for the compensating force. This is true for the crystals where the orientation structure is simple in the continuum limit; see Appendix A.2 for a detailed analysis. We have also tried modifying the external force, and the crystals found in the simulations are to large extent insensitive to such modifications. Moreover, convergence to crystals of good enough quality for our purposes typically requires adding a pseudo-temperature in the form of random kicks to the instantons that would slow down the relaxation process. The setup for the simulations is discussed in more detail in Appendix B.

Notice that the simulation results contain finite size effects which may not be highly suppressed even for 10000 instantons. In particular, compared to simulations in lower dimensions [15, 16], the simulation volume is effectively smaller compared to the boundary: For example, the distance of a random instanton to the boundary is $\sim N^{1/d}$ where $d$ is the dimension, and is reduced as $d$ increases. This enhances the dependence on boundary conditions and other boundary effects. This is seen in the simulation results, e.g., as the emergence of unwanted layer structures which are aligned with the surface of the volume of the simulation, and only appear near the surface. Therefore it is important to check that the results are independent of the number of instantons included in the simulation. Moreover the choice of initial conditions (even if they are random) as well as the pseudo-temperature may cause bias towards some of the crystals we find in the results. It is also important to compare the results with those obtained by using other methods in sections 4 and 5. We typically find good agreement, as we will discuss below.

Originally, we studied the standard unoriented case, given by the potential (7), and the standard instanton interaction, given in (5). Then it is natural to also consider the interaction that interpolates between these two, i.e., the choices with $\alpha = \beta$ in the more general interaction of (9). However, as it turns out, the line $\alpha = \beta$ is a critical line, that is, a phase boundary on the $(\alpha, \beta)$ phase diagram determined by the general interactions. Therefore we decided that it is best to extend the studies to the whole $(\alpha, \beta)$ plane.

There are however some quite obvious constraints on the values of $\alpha$ and $\beta$. Namely, for a pair $(n, m)$ of instantons there are orientation choices where the factors $\mathrm{tr}^2(\cdots)$ in (9) take independently any possible value, i.e., a value within the range from zero to four. That is, if $\alpha < -1/8$ or $\beta < -1/8$ the factor in the square brackets can become negative, so that the interaction turns attractive (at all distances). Such an interaction will lead to a collapse of the system, and therefore must be excluded from the phase diagram. That is, we restrict our studies to the region $\alpha > -1/8$ and $\beta > -1/8$. In addition, we have decided to exclude from our analysis the potentials where $\alpha$ or $\beta$ is much larger than one, so that the orientation dependent terms would dominate the interaction.

We will then discuss our simulation results. We start by the unoriented and oriented standard cases which are given by the points $\alpha = 0 = \beta$ and $\alpha = 1 = \beta$ on the phase diagram, respectively. Then we will describe the results in the rest of the plane, taking into account the constraints discussed above.

## 3.1 Unoriented

Let us start from the unoriented potential, given in (7). The convergence of the simulations is far from optimal for the unoriented potential: we obtain fcc structures, but domains are very small. However clear domains of any other regular structures are rare. In some instances, chunks of crystal closer to bcc than to fcc were found. This suggests that fcc and bcc are very close in energy, which may explain in part why it is difficult to identify the lattice with lowest energy. We will confirm this in Sec. 5, and show that the fcc structure indeed has the lowest energy. A sample of a simulation result is shown in fig. 1. This sample is a carefully chosen region of a simulation with 10000 instantons. It represents roughly the cleanest fcc structure that we could obtain. Notice that this figure, as well as the other figures in this section, may be viewed as an interactive 3D version if the pdf is opened in Adobe Reader or Adobe Acrobat (version 9 or higher).

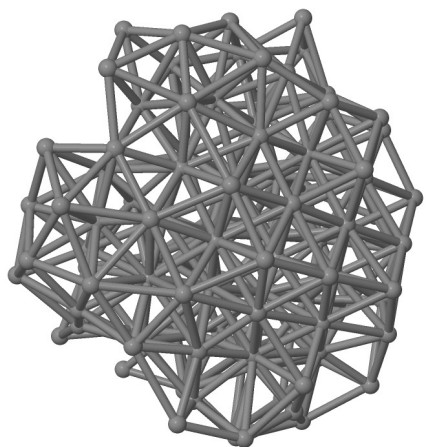

Figure 1: A sample of a simulation result for the unoriented two-body interaction showing signs of fcc structure. The figure was created by using the Jmol software [45]. An interactive 3D version of the figure can be viewed with Adobe Reader.

## 3.2 Oriented "standard" instanton interactions ($\alpha = \beta = 1$)

Let us then discuss our results for the original interaction (5). In this case, the crystal with lowest energy could be identified with high certainty, and the result is somewhat surprising. Namely, the spatial structure is a face-centered tetragonal lattice with a fairly large aspect ratio, i.e., fcc with one direction rescaled, breaking the cubic symmetry. The lattice can be equally described as body-centered tetragonal, which is related to the face centered version by a rescaling of the aspect ratio by $\sqrt{2}$ (and a rotation by 45 degrees of the unit cell). The aspect ratio $c$ is indeed large: we find[2] $c \approx 2.35$ (using the face-centered conventions). That is, the instantons form clearly separated layers with two-dimensional square lattice structure. A sample of this structure is shown in fig. 2. This sample was taken from a well-converged simulation with 5000 instantons.

The orientation structure could also be extracted reliably. The two dimensional layers have anti-ferromagnetic structure. The orientations between the layers repeat in cycles of two, as seen from fig. 2. If we choose the unit cell of the lattice to be aligned with the axes such that $z$ is the asymmetric direction, the orientations can be chosen to be 1 and $k$ in one set of the layers, whereas the other set has the directions $i$ and $j$ (where we used the standard $(1, i, j, k)$ basis for the quaternions). Recall that the orientations are determined only up to signs and left multiplication by a constant unimodular quaternion. These four orientations are shown as different colors in fig. 2.

Finally, we remark that the standard instanton interactions lie on a special line of the phase diagram of general interactions, which depends in the parameters $\alpha$ and $\beta$. That is, there is a phase transition on the line $\alpha = \beta$ on the $(\alpha, \beta)$-plane, and above this line the structure is drastically different (as we will detail below) from the result of fig. 2 while below the line they are similar. The fact that the results for $\alpha = \beta$ are continuously connected to the lower phase may be dependent on finite size effects and other details of the simulation (e.g., the pseudo-temperature). Actually in simulations we find that the critical line is at small positive $\beta - \alpha$. Despite the point being on the critical line, the convergence on this line is much faster than for

---

[2]This estimate, which is based on the simulation results, turns out to be a bit below the true value, see Sec. 5 below.

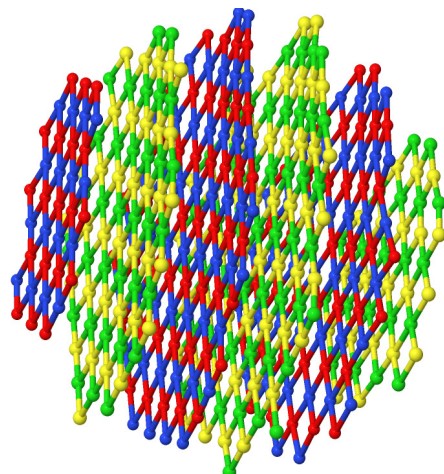

Figure 2: A sample of a simulation result for the oriented two-body interaction of 5d instanton dynamics (i.e. with $\alpha = \beta = 1$). The lattice is tetragonal with non-Abelian orientation structure. The colors show the four different orientations of the instantons. The figure was created by using the Jmol software [45]. An interactive 3D version of the figure can be viewed with Adobe Reader.

the unoriented case and the resulting crystal have typically good quality.

## 3.3 Oriented non-Abelian crystals ($\alpha \geq \beta$)

Let us then consider the simulation results for the generalized interaction (9), varying values of the parameters $\alpha$ and $\beta$. We will here only sketch the phase diagram based on the simulation results, and the final result for the diagram will be presented in Secs. 4 and (5) by using the other methods. The orientation of fig. 2 structure falls in a wider class which we call "non-Abelian" crystals. In this class, there is no two-dimensional subspace which would contain all the instanton orientations. The set as such does not single out any preferred direction. We start by exploring the region where non-Abelian crystals are found. Based on the simulation results, this means that $\alpha \gtrsim \beta$, $\alpha \gtrsim 0$, and $\beta > -1/8$.

The phase of the non-Abelian crystals may be further divided into smaller subphases. We observe that the simulation results show most variability near "critical" lines given by $\alpha = \beta$, $\beta = 0$, and $\beta = -1/8$. Outside these critical regions, we find plain fcc when $\beta > 0$ and plain simple cubic when $\beta < 0$. Examples of simulation results for both of these lattices are shown in fig. 3. The orientation pattern for the fcc lattice is the natural one with four orientations: it is actually the same as in fig. 2, but just the aspect ratio is much smaller, equal to one in this case. For the cubic lattice the orientation pattern is also the natural pattern. That is, for both these lattices the way the orientations are distributed on the instanton lattice does not single out any direction, so that the symmetry of the lattice protects the value of the aspect ratios (which equal one for both lattices). In other words, one can choose a set of three orthogonal axes such that the lattice is invariant under 90 degree rotations around all the axes.

Near the critical lines, however, other kind of lattices are found. We first discuss the line with $\beta = \alpha$ which has the richest phase diagram. As we remarked above, this line is actually the boundary of the non-Abelian phase, but due to finite size (or other simulation) effects our simulations converge to non-Abelian crystals also on this line.

For $0.5 \lesssim \alpha < 1$ the structure is the tetragonal lattice described in Sec. 3.2 and shown in

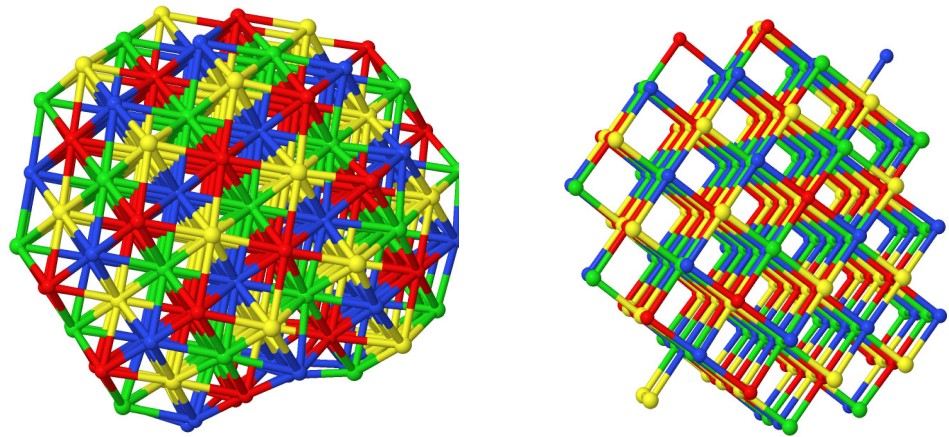

Figure 3: Generic lattice structures found in the non-Abelian phase. Left: A sample of fcc lattice from a simulation with 5000 instantons at $\alpha = 1$ and $\beta = 0.5$. Right: A sample of simple cubic lattice from a simulation with 6000 instantons at $\alpha = 0.5$ and $\beta = -0.03$.

fig. 2, except that the aspect ratio slightly decreases with decreasing $\alpha$. At $\beta = \alpha \approx 0.5$ we encounter a first order phase transition to another kind of lattice, which appears to be dominant for $0.2 \lesssim \alpha \lesssim 0.5$. This is a hexagonal lattice, which has the hexagonal close packed (hcp) structure but smaller aspect ratio (in the direction perpendicular to the 2D layers with triangular structure). There are in total six different orientations, which alternate between the 2D triangular layers. Let us take the layers to be aligned with the $xy$ plane and separated in the $z$ direction. Then one set of the triangular layers has orientations in the plane spanned by 1 and $k$, which are related to each other by rotations by 120 degrees. The layers with different orientation structure have orientations in the plane spanned by $i$ and $j$, similarly related through rotations by 120 degrees in the plane. We give the precise description of the structure in Sec. 5. A sample of this lattice is shown in fig. 4 (top left), which is taken from a simulation with 3000 instantons. Notice that for this lattice, the simulation result is particularly clean for the spatial structure (while there are some minor defects in the orientations). We used a coloring scheme which is different from the other plots to show all the six orientations.

At $\beta = \alpha \approx 0.2$ the simulations suggest that there is another phase transition, interestingly, to another fcc related structure with different value for the aspect ratio, which is now $c \sim 0.7$. That is, the lattice is close to bcc. A sample of this lattice is shown in fig. 4 (top right), and it is related to those in fig. 2 and fig. 3 (left) by rescaling of one of the coordinates.

We have also checked what happens for $\beta = \alpha > 1$. Simulations indicate that there is at least one phase transition: instead of layers with anti-ferromagnetic 2D square lattices, one obtains layers of anti-ferromagnetic hexagonal lattices at high values of $\alpha$, see fig. 4 (bottom left). It is difficult to pinpoint the value of the phase transition precisely based on the simulation results, which often have domains of both lattices, but it appears to be near $\alpha = \beta = 2$.

Let us then discuss the simulation results near $\beta = 0$. We encounter another structure, which is shown in fig. 4 (bottom right). The spatial structure is close to bcc for $\beta = 0$ exactly, but the orientations are arranged in a different way than in fig. 4 (top right). When $\beta \neq 0$ but small, we observe that the aspect ratio varies relatively fast (increasing with $\beta$).

The remaining critical line is $\beta = -1/8$. Our simulations in this region converge slowly, but suggest the existence of simple tetragonal phases, i.e., lattices of fig. 3 (right), but with

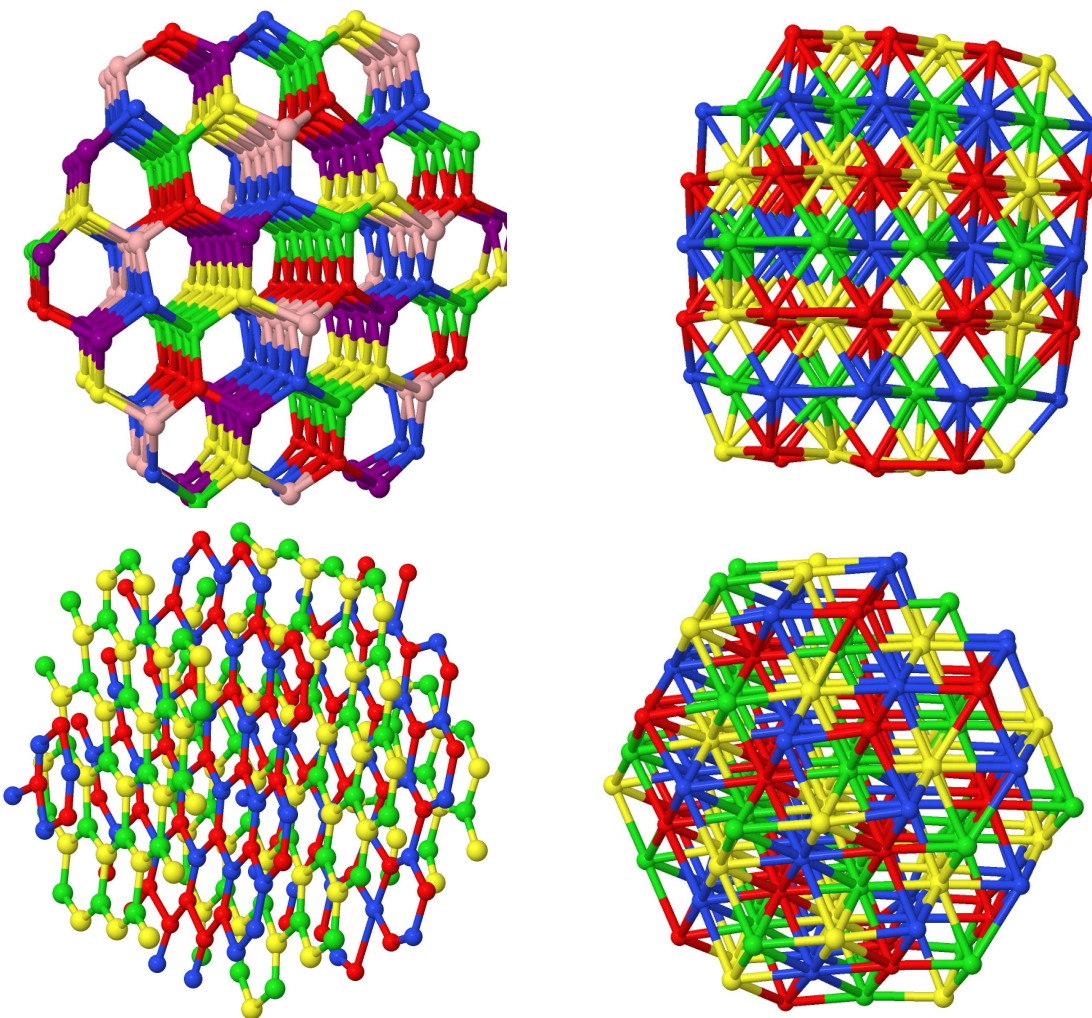

Figure 4: Additional lattice structures found for $0 < \alpha = \beta < 1$ and for $\beta = 0$. Top left: A sample of hexagonal lattice from a simulation with 3000 instantons at $\alpha = \beta = 0.4$. Top right: A sample of a (nearly) bcc lattice from a simulation with 3000 instantons at $\alpha = \beta = 0.07$. Bottom left: A sample of a lattice with hexagonal antiferromagnetic layers from a simulation with 5000 instantons at $\alpha = 10$ and $\beta = 10$. Bottom right: A sample of a bcc lattice (with nonstandard orientation structure) from a simulation with 5000 instantons at $\alpha = 0.6$ and $\beta = 0$.

nontrivial aspect ratio. We will return to this in Sec. 5.

## 3.4 Oriented (anti) ferromagnetic spherical phase ($0 < \alpha < \beta$)

Next we discuss the results in the region $0 < \alpha < \beta$. In this region the results are drastically different from those at $\alpha > \beta$: there is a very strong tendency for the orientations to align with the position of the instanton, which leads to a global spherical orientation structure. We note that simulations in these "spherical" phases converge to the final state over ten times faster than for the non-Abelian phases found for $\beta \leq \alpha$ which suggests that the spherically arranged ground states are energetically strongly favored. The simulation results agree with analytic analysis of

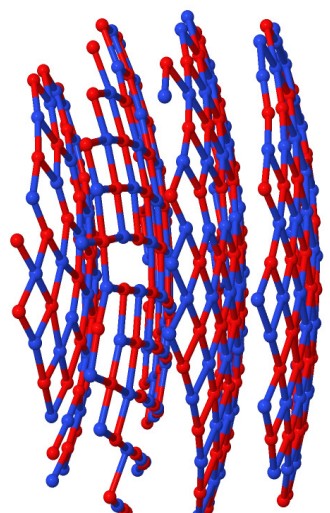
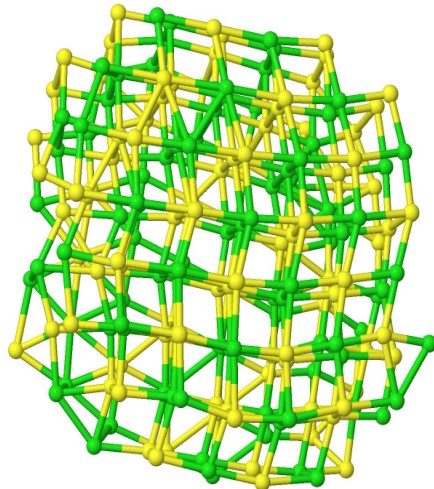

Figure 5: Anti-ferromagnetic spherical lattices found for $0 < \alpha < \beta$. Left: A sample with two-dimensional square lattice layers from a simulation with 15000 instantons at $\alpha = 2/3$ and $\beta = 1$. Right: A sample with two-dimensional rhombic layers from a simulation with 15000 instantons at $\alpha = 0.1$ and $\beta = 0.15$.

the long distance behavior we will carry out in the next section: in all simulations for $0 < \alpha < \beta$, we find that the orientations are arranged spherically. The spherical phase further divides into two regions, where the orientation structure is either locally antiferromagnetic (constraining to small regions, two orientations appear, with closest distance links being between different orientations) or ferromagnetic (in small regions, all orientations are aligned). We verify that there is a phase transition from (local) anti-ferromagnetic to (local) ferromagnetic order at $\beta = \Gamma\alpha$ with the value of $\Gamma$ between 2.2 and 2.3. This is close to the analytic prediction $\Gamma \approx 2.14$ from Sec. 4, and the small deviation is probably a finite size effect and/or a bias due to the simulation setup.

In the anti-ferromagnetic phase we observe a very strong tendency of the system to converge to almost exactly spherical shells. While there is some tendency to form spherical shells also in the other phases, at least near the boundary of the spherical cavity in our simulations, in the anti-ferromagnetic phase this effect is clearly stronger than in the other phases. That is, it is plausible that the spherical shells observed in the other phases are essentially boundary defects, but in the anti-ferromagnetic phase, even for the largest simulations with 40000 instantons which we have run, the whole system converges rapidly to easily recognizable spherical structures.

Within the anti-ferromagnetic phase, the simulation result also strongly depends on $\alpha$ and $\beta$. For large values, $\alpha^2 + \beta^2 \gtrsim 0.5^2$, the spherical layers are clearly separated and form two-dimensional anti-ferromagnetic square lattices, see fig. 5 (left) for an example. Due to finite size effects it is difficult to pin down the preferred local crystal structure, but we conjecture that it is simple tetragonal so that the different layers prefer to be precisely aligned with exactly same orientation pattern. The separation between the layers slightly grows with $\alpha$ and $\beta$. Closer to the origin on the $(\alpha, \beta)$-plane, the layers are rhombic instead of exactly rectangular, see fig. 5 (right). At very small values of the parameters, i.e. for $\alpha^2 + \beta^2 \lesssim 0.1^2$, the two-dimensional layers are closer to triangular, and the distance between the layers is comparable to the distances between nearest-neighbor instantons within the layers. The crystal structure is locally close to bcc, while the orientations remain anti-ferromagnetic. It is difficult to pin down the

precise preferred orientation structure in this region as the orientation dependent terms in the interaction potential are weak.

We notice that the simulation results in the anti-ferromagnetic phase therefore have some resemblance to those in the non-Abelian phase at $\alpha = \beta$: At $\alpha^2 + \beta^2 \sim 1$ we find clearly separated square lattices in both phases, and at small $\alpha^2 + \beta^2$ we find bcc-like structures in both phases. Hexagonal lattices, which are found in the non-Abelian phase, are however absent in the anti-ferromagnetic phase.

Finally, we have also carried out simulations in the ferromagnetic phase ($\beta > \Gamma\alpha > 0$). As we pointed out above, the results are in agreement with the analytic long distance analysis: the orientations are spherically aligned. There is some tendency of forming spherical structures but this is considerably weaker than in the anti-ferromagnetic phase. It is again difficult to determine the local crystal structure with high certainty but it is plausible that it is fcc: locally the simulation results are similar to fig. 1. The simulation results are essentially independent of $\alpha$ and $\beta$ within this phase.

Notice also that, as we remarked above, for the spherical crystals it is not obvious what one should choose as the compensating external force (see detailed discussion in Appendix A.2). In order to make sure that our results are consistent, we have also run simulations for different choices of potentials. For example, instead of the potential which enforces constant instanton density, we have used potentials which are due to an IR regulated potential obtained by integrating a continuum of spherically arranged instantons outside the cavity of simulation. Changes in the external force did not lead to any changes in the local crystal structure, or remove the global spherical arrangement of the orientations. Furthermore, since the spherical structures are surprising, we have checked that the results are insensitive to the boundary conditions by varying the size of the ensemble and the shape of the cavity of simulation. We were able to carry out simulations with up to about 40000 instantons without any essential change in the results. Naturally, we cannot exclude that some changes occur for even larger ensembles.

## 3.5 Global ferromagnetic phases ($\alpha < 0$)

The line of $\alpha = 0$ marks another phase transition. Namely, for negative $\alpha$ (and with $\alpha > -1/8$), we find a simple global ferromagnetic structure, where all instantons have the same orientation to a good precision. The two-body potential is then proportional to that of the same lattice is then fcc, and simulation results are similar than that shown in fig. 1.

There is however a small corner in the region where both $\alpha$ and $\beta$ which shows highly unexpected behavior: the orientation structure is not uniform. The region where this happens is given by $0.5\beta \lesssim \alpha \lesssim 0$ and $\beta > -1/8$. Most of the simulation result obeys a global ferromagnetic structure, but there is a crust that is locally ferromagnetic but globally spherical. The crust is relatively thin (typically having thickness of about 10% of the radius of the simulation) but it scales with the system size and therefore is present even in the continuum limit. We discuss this structure in more detail in the next section.

## 4 The instanton energy at long distances

In this section we calculate the net self-interaction energy in a large spherical piece (radius $R$) of the instanton matter of density $\rho$. Due to linear IR divergence of $1/\text{distance}^2$ two-body potential in 3D, the net IR energy — which is independent on the instanton lattice geometry — scales like $E_{\text{IR}} \sim \rho^2 R^4$, much larger than the geometry-dependent effect whose net contribution scales like

$\Delta E \sim \rho^2 R^3 a$ where $a$ is the lattice spacing. In this section we focus on the leading IR effects only, so we take the continuum limit $a \to 0$ and ignore the lattice geometry — it can be cubic, hexagonal, tetragonal, whatever, it does not matter.

However, unlike the lattice geometry, the orientation pattern of the instantons does affect the IR energy of the lattice. Or rather, what matters is the spectrum of instanton orientations in each small but macroscopic volume $d^3\mathbf{x}$ of the instanton matter; but the IR limit does not care for the specific locations of those orientations within the instanton lattice. The spectrum of orientations matters because the two-body force between the instantons depends on their orientations. For convenience we set the prefactor of eq. (1) to one.

$$\frac{2N_c}{\lambda M(1 + 2\alpha + 2\beta)} \to 1 \,, \tag{14}$$

so we have

$$V_{1,2} = \frac{Q_{12}}{|\mathbf{x}_1 - \mathbf{x}_2|^2} \quad \text{for} \quad Q_{12} = \frac{1}{2} + \alpha \mathrm{tr}^2(y_1 y_2^\dagger) + \beta \mathrm{tr}^2((i\mathbf{n}_{12} \cdot \vec{\tau}) y_1 y_2^\dagger) \,, \tag{15}$$

where $y_1$ and $y_2$ are the two instantons' orientations in $SU(2)$ notations and $\mathbf{n}_{12}$ is the unit vector in the direction of $\mathbf{x}_2 - \mathbf{x}_1$. For our purposes, we treat the two instantons' locations $\mathbf{x}_1$ and $\mathbf{x}_2$ as continuous variables, and *average* the $Q_{12}$ over the orientations of instantons found within small but macroscopic volumes $d^3\mathbf{x}_1$ and $d^3\mathbf{x}_2$. Given such an average $\langle Q_{12} \rangle$, the net IR energy of the instanton matter of *macroscopic* density $\rho(\mathbf{x})$ is

$$E_{\mathrm{IR}} = \frac{1}{2} \int d^3\mathbf{x}_1 \, d^3\mathbf{x}_2 \, \frac{\rho(\mathbf{x}_1)\rho(\mathbf{x}_2)\langle Q_{12} \rangle}{|\mathbf{x}_1 - \mathbf{x}_2|^2} \,. \tag{16}$$

Since our numeric simulations used instanton matter of approximately uniform density confined to a spherical cavity of large radius $R$, we are going to use similar setting in this section: Uniform $\rho(\mathbf{x}) = \text{const}$ while all space integrals are limited to a ball of radius $R$. Consequently, the net IR energy of the instanton matter becomes

$$E_{\mathrm{IR}} = \frac{\rho^2}{2} \int_{\mathrm{ball}} d^3\mathbf{x}_1 \, d^3\mathbf{x}_2 \, \frac{\langle Q_{12} \rangle}{|\mathbf{x}_1 - \mathbf{x}_2|^2} \,. \tag{17}$$

For future reference, let us explicitly calculate the IR energy of the baseline case of constant $\langle Q_{12} \rangle = 1$:

$$E_{\mathrm{base}} = \frac{\rho^2}{2} \int_{\mathrm{ball}} \frac{d^3\mathbf{x}_1 \, d^3\mathbf{x}_2}{|\mathbf{x}_1 - \mathbf{x}_2|^2} = 2\pi^2 \rho^2 R^4 \,. \tag{18}$$

This baseline energy serves as a convenient unit of IR energies for various orientation phases of the instanton matter. In particular, for the phases having uniform $\langle Q_{12} \rangle = \text{const}$, we immediately have

$$E_{\mathrm{IR}} = \langle Q_{12} \rangle \times E_{\mathrm{base}} \,. \tag{19}$$

Through the rest of this section, we shall calculate the $\langle Q_{12} \rangle$ as a function of $\mathbf{x}_1$ and $\mathbf{x}_2$ and hence the net IR energy for all the orientation patterns we have observed in our simulations, namely the non-Abelian patterns, the spherical ferromagnetic and antiferromagnetic patterns, the global ferromagnetic pattern, and the ferromagnetic 'egg' (which obtains for some negative $\alpha$ and $\beta$). For completeness sake, we also include the global antiferromagnetic pattern we have not observed but suspected might show up in some obscure corner of parameter space.

## 4.1 Non-Abelian patterns

Let's start with the non-Abelian patterns involving 4, 6 or perhaps more different instanton orientations in any lattice cell. In terms of the $S^3$ group manifold of the $SU(2)$, the net 4D quadrupole momentum of such orientations should have no traceless part. In the 4-orientation case $y = 1, i\tau_1, i\tau_2, i\tau_3$ (so that the corresponding quaternions are $1, i, j, k$) vanishing of the traceless quadrupole moment is obvious; we have checked that it also vanishes for the 6-orientation pattern of a hexagonal lattice and we believe this is a general case for any non-Abelian orientation pattern. In terms of the instanton orientations $y_i$ themselves, this means that averaging over a complete periodic cell of the lattice — and hence over any macroscopic volume — one gets

$$\text{average over } y_i \text{ of } \text{tr}^2(y_i Y^\dagger) \text{ is } 1 \quad \text{for any given } SU(2) \text{ matrix } Y. \tag{20}$$

In the context of (9), $Y$ can be either $y'$ of a distant instanton or $(i\mathbf{n} \cdot \tau)y'$, hence

$$\langle Q_{12} \rangle = \frac{1}{2} + \alpha + \beta.$$

Since this averaged coefficient of the $1/r^2$ force is constant, it follows that the net IR energy is

$$E_{\text{IR}}^{\text{NA}} = \left( \frac{1}{2} + \alpha + \beta \right) \times E_{\text{base}}. \tag{21}$$

## 4.2 Global ferromagnetic pattern

Now consider the global ferromagnetic pattern in which all the instantons have the same orientation $y = \text{const}$. Let's assume no macroscopic domains, so that $y$ stays the same over the whole big ball of the instanton matter. In this case $y_1 y_2^\dagger = 1$ for any two instantons, hence

$$\langle Q_{12} \rangle = \frac{1}{2} + 4\alpha.$$

Again, we have a constant $\langle Q \rangle$ (although a different constant from the NA patterns), hence the net IR energy is

$$E_{\text{IR}}^{\text{FM}} = \left( \frac{1}{2} + 4\alpha \right) \times E_{\text{base}}. \tag{22}$$

## 4.3 Global anti-ferromagnetic pattern

Next, the global anti-ferromagnetic pattern in which instantons have 2 alternating orientations $y_a$ and $y_b$ related by (the $SU(2)$ representation of) a 180° rotation around some axis $\mathbf{N}$, that is $y_b = \pm i(\mathbf{N} \cdot \vec{\tau}) y_a$. By the *global* AF pattern we mean there are no domains so the two orientations $y_a$ and $y_b$ are the same everywhere in the big ball of the instanton matter, thus

$$y_1 y_2^\dagger = \begin{cases} \text{either} & 1 \\ \text{or} & i\mathbf{N} \cdot \tau \end{cases}. \tag{23}$$

Averaging between these two cases, we get

$$\langle Q_{12} \rangle = \frac{1}{2} + 2\alpha + 2\beta(\mathbf{N} \cdot \mathbf{n}_{12})^2, \tag{24}$$

where $\mathbf{n}_{12}$ is the direction of the line between the two instantons. For the purpose of calculating the net IR energy, we may further average the $\langle Q \rangle$ over the directions of the $\mathbf{x}_2$ and of the $\mathbf{x}_1$ and hence over the directions of the $\mathbf{n}_{12}$, thus

$$\langle\langle Q_{12}\rangle\rangle = \frac{1}{2} + 2\alpha + 2\beta \times \frac{1}{3} = \text{const}, \tag{25}$$

and therefore

$$E_{\text{IR}}^{\text{AF}} = \left(\frac{1}{2} + 2\alpha + \frac{2}{3}\beta\right) \times E_{\text{base}}. \tag{26}$$

## 4.4 Spherical anti-ferromagnetic pattern

Now consider the anti-ferromagnetic spherical shells that we found in the simulations for $\beta > \alpha$. In the anti-ferromagnetic case, the microscopic orientation pattern is anti-ferromagnetic, but macroscopically the alternate orientations $y_a$ and $y_b$ vary from place to place such that near $\mathbf{x}$ the flip axis $\mathbf{N}$ points in the direction of $\mathbf{x}$, thus

$$y_b(\mathbf{x}) = \pm i(\mathbf{n}_x \cdot \tau) y_a(\mathbf{x}). \tag{27}$$

Note: taking 2 lattice steps in some direction, an AF orientation $y_a$ should turn into the $y_b$ and then back into $y_a$, but since the $\mathbf{n}_x$ unit vectors for the two steps are slightly different, we find that the $y_a$ orientations at the double-step separation are not exactly the same. Taking the continuum limit, we get a differential equation for the $y_a(\mathbf{x})$, namely

$$dy_a(\mathbf{x}) = \left(i\frac{d\mathbf{x} \times \mathbf{x}}{2|\mathbf{x}|^2} \cdot \vec{\tau}\right) y_a(\mathbf{x}). \tag{28}$$

This equation has a unique solution up to a global $SU(2)$ rotation (which is irrelevant to our purposes). In polar coordinates

$$
\begin{aligned}
y_a(r,\theta,\phi) &= \cos(\theta/2) + i\sin(\theta/2)\big(\sin\phi\,\tau_1 - \cos\phi\,\tau_2\big) \\
&= \begin{pmatrix} +\cos(\theta/2) & -\sin(\theta/2)e^{-i\phi} \\ +\sin(\theta/2)e^{+i\phi} & +\cos(\theta/2) \end{pmatrix}, \\
y_b(r,\theta,\phi) &= i\cos(\theta/2)\tau_3 + i\sin(\theta/2)\big(\cos\phi\,\tau_1 + \sin\phi\,\tau_2\big) \\
&= \begin{pmatrix} +i\cos(\theta/2) & +i\sin(\theta/2)e^{-i\phi} \\ +i\sin(\theta/2)e^{+i\phi} & -i\cos(\theta/2) \end{pmatrix}.
\end{aligned} \tag{29}
$$

For the position-dependent $y_a(\mathbf{x})$ and $y_b(\mathbf{x})$ like this, averaging $Q_{12}$ means averaging over $y_1 = y_a(\mathbf{x}_1)$ or $y_b(\mathbf{x}_2)$ as well as over $y_2 = y_a(\mathbf{x}_2)$ or $y_b(\mathbf{x}_2)$. This takes Mathematica and a bit of work, so let us simply quote the result:

$$\langle Q_{12}\rangle = \frac{1}{2} + \alpha\big(1 + (\mathbf{n}_1 \cdot \mathbf{n}_2)\big) + \beta\big(1 - (\mathbf{n}_1 \cdot \mathbf{n}_2) + 2(\mathbf{n}_1 \cdot \mathbf{n}_{12})(\mathbf{n}_2 \cdot \mathbf{n}_{12})\big). \tag{30}$$

Or in terms of the radii $r_1 = |\mathbf{x}_1|$, $r_2 = |\mathbf{x}_2|$ and the angle $\theta_{12}$ between the $\mathbf{x}_1$ and $\mathbf{x}_2$ directions,

$$\langle Q_{12}\rangle = (\frac{1}{2} + \alpha + \beta) + (\alpha - \beta) \times \cos\theta_{12} - 2\beta \times \frac{(r_1 - r_2\cos\theta_{12})(r_2 - r_1\cos\theta_{12})}{r_1^2 + r_2^2 - 2r_1r_2\cos\theta_{12}}. \tag{31}$$

Consequently, the net IR energy of the instanton matter with this orientation pattern is

$$E_{\text{SAF}}^{\text{IR}} = \left(\frac{1}{2} + \alpha + \beta\right) \times E_{\text{base}} + (\alpha - \beta) \times \Delta_1 E - 2\beta \times \Delta_2 E, \tag{32}$$

where

$$\Delta_1 E = \frac{\rho^2}{2} \int_{\text{ball}} d^3\mathbf{x}_1 \, d^3\mathbf{x}_2 \frac{\cos\theta_{12}}{|\mathbf{x}_1 - \mathbf{x}_2|^2} \Delta_1 V(\mathbf{x}_1) = 2\pi^2 \rho^2 R^4 \times \frac{4\log(2) - 1}{3}$$

$$= E_{\text{base}} \times \frac{4\log(2) - 1}{3} \approx 0.59 E_{\text{base}}, \tag{33}$$

while

$$\Delta_2 E = \frac{\rho^2}{2} \int_{\text{ball}} d^3\mathbf{x}_1 \, d^3\mathbf{x}_2 \frac{(r_1 - r_2\cos\theta_{12})(r_2 - r_1\cos\theta_{12})}{|\mathbf{x}_1 - \mathbf{x}_2|^4} = 0. \tag{34}$$

Altogether,

$$E_{\text{SAF}}^{\text{IR}} = \left(\frac{1}{2} + (1 + \nu)\alpha + (1 - \nu)\beta\right) \times E_{\text{base}} \tag{35}$$

$$\text{where } \nu = \frac{4\log(2) - 1}{3} \approx 0.59. \tag{36}$$

## 4.5 Spherical ferromagnetic pattern

Finally, consider the ferromagnetic spherical shell pattern in which the orientation is locally ferromagnetic — all the instantons in a neighborhood have similar orientation $y$, — but globally $y(\mathbf{x})$ slowly varies from place to place. In particular, in the pattern seen in the simulations,

$$y(\mathbf{x}) = \pm i\mathbf{n}_x \cdot \tau, \tag{37}$$

up to a global $SU(2)$ rotation. For this pattern, we need no averaging but $\langle Q_{12} \rangle = Q_{12}$ depends on the angle $\theta_{12}$ between the directions of the instantons' locations $\mathbf{x}_1$ and $\mathbf{x}_2$. Specifically,

$$\text{tr}(y(\mathbf{x}_1)y^\dagger(\mathbf{x}_2)) = \pm 2\cos\theta_{12}, \tag{38}$$

while

$$\text{tr}\big(y(\mathbf{x}_1)y^\dagger(\mathbf{x}_2)(i\mathbf{n}_{12}\tau)\big) = \pm 2(\mathbf{n}_1 \times \mathbf{n}_2) \cdot \mathbf{n}_{12} = 0, \tag{39}$$

hence

$$\langle Q_{12} \rangle = \frac{1}{2} + 4\alpha\cos^2\theta_{12}. \tag{40}$$

The net IR energy is rather simple:

$$E_{\text{SFM}}^{\text{IR}} = \frac{\rho^2}{2} \int_{\text{big ball}} d^3\mathbf{x}_1 d^3\mathbf{x}_2 \frac{\frac{1}{2} + 4\alpha\cos^2\theta_{12}}{|\mathbf{x}_1 - \mathbf{x}_2|^2}$$

$$= \pi^2 \rho^2 R^4 \times \left(1 + \frac{\pi^2}{2}\alpha\right) = E_{\text{base}} \times \left(\frac{1}{2} + \frac{\pi^2}{4}\alpha\right). \tag{41}$$

## 4.6 Summary of homogeneous orientation patterns

Thus far, we have calculated the IR energy of all the homogeneous orientation patters we have seen in our simulations:

any non-abelian pattern:

$$E_{\text{NA}} = \left(\tfrac{1}{2} + \alpha + \beta\right) \times E_{\text{base}}, \tag{21}$$

global ferromagnetic pattern:

$$E_{\text{FM}} = \left(\tfrac{1}{2} + 4\alpha\right) \times E_{\text{base}}, \tag{22}$$

global anti-ferromagnetic pattern:

$$E_{\text{AF}} = \left(\tfrac{1}{2} + 2\alpha + \tfrac{2}{3}\beta\right) \times E_{\text{base}}, \tag{26}$$

spherical anti-ferromagnetic shells:

$$E_{\text{SAF}} = \left(\tfrac{1}{2} + (1+\nu)\alpha + (1-\nu)\beta\right) \times E_{\text{base}} \tag{35}$$
$$\text{for } \nu = \frac{4\log(2)-1}{3} \approx 0.59,$$

spherical ferromagnetic shells:

$$E_{\text{SFM}} = \left(\frac{1}{2} + \frac{\pi^2}{4}\alpha\right) \times E_{\text{base}}. \tag{41}$$

Actually, we have never seen the global anti-ferromagnetic pattern in our simulations, but we have included it just in case.

Now let's compare these IR energies for various ranges of $\alpha$ and $\beta$ and find which pattern has the lowest IR energy. Note however that the UV stability of the instanton matter requires the two-body $1/r^2$ to be repulsive for all orientations and hence $\alpha \geq -\tfrac{1}{8}$ and $\beta \geq \tfrac{1}{8}$. Also, in this section we shall focus on the parts of the parameter space where $\alpha \geq 0$, or $\beta \geq 0$, or both. The remaining part of the parameter space where both $\alpha$ and $\beta$ are negative will be explored in the next subsection 4.7 once we calculate the energy of the in-homogeneous ferromagnetic egg pattern.

Here is the summary of our results:

- The non-Abelian orientation patterns win — *i.e.* have lower IR energy that any other pattern — for $\alpha > \beta > 0$ and also for $\alpha > 0$ while $\beta < 0$ (but $\beta \geq -\tfrac{1}{8}$).

  - Note that there are several different non-Abelian patterns, but they all have degenerate IR energies. In the next section 5 we shall compare their energies beyond the IR limit and find the specific winning non-Abelian patterns for each range of $\alpha$ and $\beta$.

- The spherical anti-ferromagnetic shell pattern wins for $\alpha > 0$ and $\beta > \alpha$ but $\beta < \Gamma\alpha$ for

$$\Gamma = \frac{\frac{\pi^2}{4} - 1 - \nu}{1 - \nu} \approx 2.14. \tag{42}$$

- The spherical ferromagnetic shell pattern wins for $\alpha > 0$ and $\beta > \Gamma\alpha$.

- The global ferromagnetic pattern wins for $\alpha < 0$ while $\beta > 0$.

× Finally, the global anti-ferromagnetic pattern never wins: for any allowed $\alpha$ and $\beta$ some other pattern has lower IR energy than the global AF patter.

To conclude this summary, let me emphasize the phase boundaries between the different IR patterns:

* The boundary between the non-Abelian and the spherical antiferromagnetic patterns lies along the positive $\beta = \alpha$ line. In our simulations, this boundary seems to lie at small but positive $\beta - \alpha$, but this is probably due to finite-sample-size or finite-temperature effects.

* The boundary between the spherical anti-ferromagnetic and ferromagnetic shells lies at positive $\beta = \Gamma\alpha$ line. In simulations, this boundary seems to lie at slightly higher $\beta/\alpha$ ratios between 2.2 and 2.3, but again this is probably due to finite-sample-size or finite-temperature effects.

* Finally, the boundary between the global and the spherical-shell ferromagnetic patterns lie along the $\alpha = 0$ line, and that's exactly what we saw in our simulations.

### 4.7 Ferromagnetic egg pattern

Thus far, we have calculated the IR energies of all the homogeneous orientation patters we have seen in out simulations. However, for some values of negative $\alpha$ and negative $\beta$ the simulations produced the in-homogeneous egg-like pattern where the central part of the spherical cavity — the yolk of the egg — is a ferromagnetic lattice of constant orientation $y$, while the outer part of the cavity — the white of the egg — is made of ferromagnetic shells where the instanton orientation $y$ slowly changes with the spherical coordinates. Moreover, all the $y(\theta, \phi)$ in the 'egg's white' are perpendicular to the uniform $y$ of the 'yolk'. Specifically, the orientations are

$$
\begin{aligned}
\text{for } 0 < r < \xi R, \quad y &= 1, \\
\text{for } \xi R < r < R, \quad y &= i\mathbf{n} \cdot \vec{\tau},
\end{aligned} \tag{43}
$$

where $R$ is the cavity's radius and $\xi R$ is the boundary between the 'yolk' and the 'egg white' orientation patterns. Typically $0.73 < \xi < 0.87$, so both the 'yolk' and the 'white' patterns take a significant fraction of the overall cavity volume.

For such ferromagnetic egg pattern, the $Q_{12}$ for for two instantons at $\mathbf{x}_1$ and $\mathbf{x}_2$ does not need macroscopic averaging, but its value depends not only on the directions $\mathbf{n}_1$ and $\mathbf{n}_2$ of the two instanton positions but also on their radii $r_1$ and $r_2$:

* For $r_1, r_2 < \xi R$ — $i.e.$ when both instantons are in the egg's yolk,

$$
Q_{12} = \tfrac{1}{2} + 4\alpha, \tag{44}
$$

similar to the global ferromagnetic pattern.

* For $r_1, r_2 > \xi R$ — $i.e.$ when both instantons are in the egg's white,

$$
Q_{12} = \tfrac{1}{2} + 4\alpha \cos^2 \theta_{12}, \tag{45}
$$

similar to the ferromagnetic spherical shell pattern.

- But when one of the instanton is in the yolk while the other is in the egg's white — say, for $r_1 < \xi R < r_2$, —

$$Q_{12} = \tfrac{1}{2} + 4\beta(\mathbf{n}_2 \cdot \mathbf{n}_{12})^2 = \tfrac{1}{2} + 4\beta \times \frac{(r_2 - r_1 \cos\theta_{12})^2}{r_{12}^2 = r_1^2 + r_2^2 - 2r_1 r_2 \cos\theta_{12}}. \tag{46}$$

For simplicity, we assume that both the yolk and the white parts of the ferromagnetic egg have equal and uniform instanton densities, $\rho(\mathbf{x}) = \mathrm{const}$ throughout the cavity. Nevertheless, for the above formulae for the $\langle Q_{12} \rangle$, the energy integral

$$E_{\mathrm{egg}} = \frac{\rho^2}{2} \iint\limits_{\substack{\text{whole}\\\text{egg}}} d^3\mathbf{x}_1 \, d^3\mathbf{x}_2 \, \frac{\langle Q_{12} \rangle}{|\mathbf{x}_1 - \mathbf{x}_2|^2}, \tag{47}$$

becomes rather complicated and takes Mathematica to evaluate. Even after several rounds of simplification, we get a mess of dilogarithmic functions,

$$E_{\mathrm{egg}} = E_{\mathrm{base}} \times \left( \tfrac{1}{2} + \alpha \times f(\xi) + \beta \times g(\xi) \right) \tag{48}$$

where
$$
\begin{aligned}
f(\xi) = {} & \frac{\pi^2}{3} + \frac{48 - \pi^2}{12} \xi^4 + 2(1 - \xi^2)^2 \operatorname{ar\,tanh}(\xi) \\
& + \frac{\xi^4}{2} \left( \log(\xi) \log\frac{\xi}{1-\xi} + \mathrm{Li}_2\frac{-1}{\xi} + \mathrm{Li}_2\frac{\xi-1}{\xi} \right) \\
& + (\xi^4 - 2)\left( \mathrm{Li}_2(\xi) - \tfrac{1}{4}\mathrm{Li}_2(\xi^2) \right)
\end{aligned}
\tag{49}
$$

and
$$
\begin{aligned}
g(\xi) = {} & -\frac{\pi^2 + 16}{2} \xi^4 + \xi + 7\xi^3 - (1 - \xi^2)(1 - 3\xi^2) \operatorname{ar\,tanh}(\xi) \\
& + 4\xi^4 \mathrm{Li}_2(\xi) - \xi^4 \mathrm{Li}_2(\xi^2).
\end{aligned}
\tag{50}
$$

Note: although these formulae for the $f(\xi)$ and $g(\xi)$ are rather formidable, numerically plotting them as functions of $\xi$ shows that *for all $\xi$ between 0 and 1,*

$$2.1 \le f(\xi) \le 4 \quad \text{and} \quad 0 \le g(\xi) < 0.95, \tag{51}$$

and also

$$f(\xi) + g(\xi) > 2 \quad \text{and} \quad f(\xi) + \Gamma g(\xi) > \frac{\pi^2}{4}. \tag{52}$$

In light of these bounds, the ferromagnetic egg pattern never wins the lowest energy competitions against the homogeneous patterns for $\alpha \ge 0$ or $\beta \ge 0$ or both $\alpha, \beta \ge 0$. In these regimes, the winner is always a homogeneous patters discussed in the previous section §4.6, namely, non-Abelian, AF spherical shell, FM spherical shell, or global FM. But the regime of both $\alpha, \beta < 0$ requires a more careful consideration.

To find the winning pattern in the $\alpha, \beta < 0$ regime, we minimize the ferromagnetic egg's energy (48) as a function of the relative yolk size $\xi$ and then compare the minimum to the other two competing patterns in this regime, namely the non-Abelian pattern and the global ferromagnetic pattern. For simplicity, we first take a difference

$$
\begin{aligned}
E_{\mathrm{egg}} - E_{\mathrm{NA}} &= E_{\mathrm{base}}\Big( \alpha \times ((f(\xi) - 1) + \beta(g(\xi) - 1) \Big) \\
&= (-\beta) E_{\mathrm{base}} \times h(\xi; \alpha/\beta)
\end{aligned}
\tag{53}
$$

$$\text{for } h(\xi; \alpha/\beta) = \frac{\alpha}{\beta}(f(\xi) - 1) + g(\xi) - 1, \tag{54}$$

and then plot $h$ as a function of $\xi$ for different values $\alpha/\beta$. If the we see the whole $h(\xi)$ curve lying above the $h = 0$ axis, then the lowest-energy pattern is non-Abelian rather than the the ferromagnetic egg. On the other hand, if the curve dips below $h = 0$, then the lowest energy is some kind of a ferromagnetic pattern: Global FM if the minimum obtains for $\xi = 1$, spherical FM shells if the minimum obtains for $\xi = 0$, and FM egg if the minimum lies for $\xi$ strictly between 0 and 1.

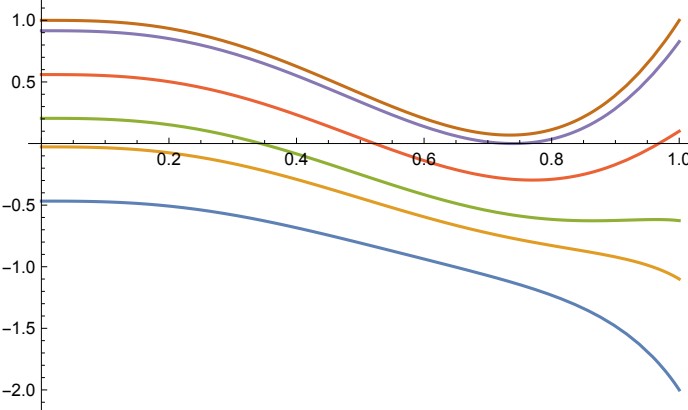

Figure 6: Plots of $h(\xi)$ — cf. eq. (53) — for several values of the $\alpha/\beta$ ratio: The blue line is for $(\alpha/\beta) = 1$, the orange line is for for $(\alpha/\beta) = 0.7$, the green line is for $(\alpha/\beta) = 0.543$ (transition), the red line is for $(\alpha/\beta) = 0.3$, the violet line is for $(\alpha/\beta) = 0.0575$ (transition), the brown line is for $(\alpha/\beta) = 0$.

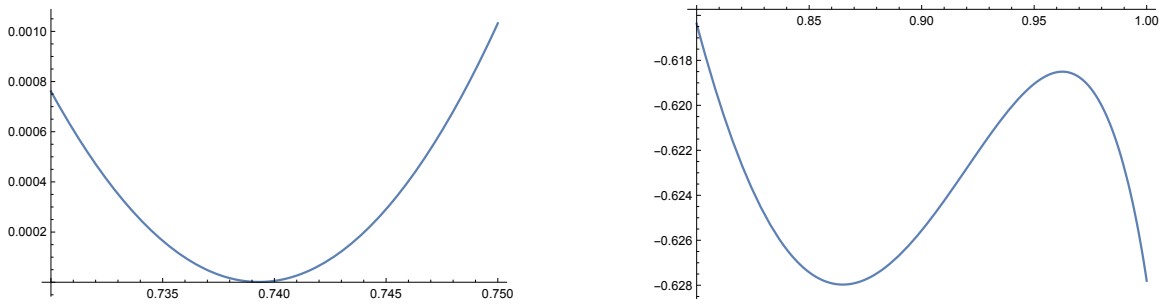

Figure 7: Zoomed-up plots of $h(\xi)$ at critical values $\alpha/\beta$ ratio. The left plot is for $(\alpha/\beta) = 0.0575$ and the right plot is for $(\alpha/\beta) = 0.543$.

Figure 6 shows the actual plots $h(\xi)$ for several values of the $\alpha/\beta$ ratio. We see that for $(\alpha/\beta) < 0.0575$ the whole curve lies above $h = 0$, so the lowest energy pattern is non-Abelian. For $(\alpha/\beta) = 0.0575$ the curve's minimum touches zero — see the left half of figure 7 for the zoomed up plot — which indicates a first-order phase transition, — and for the higher $(\alpha/\beta)$ ratios the minimum goes negative, so the lowest energy pattern is ferromagnetic. Moreover, as long as $(\alpha/\beta) \lesssim 0.5$ there is a unique minimum at $\xi < 1$ so the winning pattern is the FM egg rather than the global FM. As we increase the $\alpha/\beta$ ratio, the minimum moves to larger $\xi$, and for $(\alpha/\beta) \approx 0.5$ the curve develops two local minima, one at $\xi \approx 0.85$ (FM egg) and the other at $\xi = 1$ (global FM). At $(\alpha/\beta) = 0.543$ — see the right half of figure 7 for the zoom up, — the two minima become degenerate, which corresponds to the first-order transition from the FM

egg pattern to the global FM patter. For the higher $(\alpha/\beta) > 0.543$, the global minimum lies at $\xi = 1$ and the winning pattern is global FM.

The bottom line is: in the regime of negative $\alpha$ and negative $\beta$, specifically $-\frac{1}{8} < \alpha, \beta < 0$, there are three IR-different orientation patterns, depending on the $\alpha/\beta$ ratio:

- The globally ferromagnetic pattern for $-\frac{1}{8} < \alpha < 0.543\beta$.

- The ferromagnetic egg pattern with relative yolk radius $0.73 < \xi < 0.87$ for $0.543\beta < \alpha < 0.0575\beta$.

- The non-Abelian pattern for $\alpha > 0.0575\beta$.

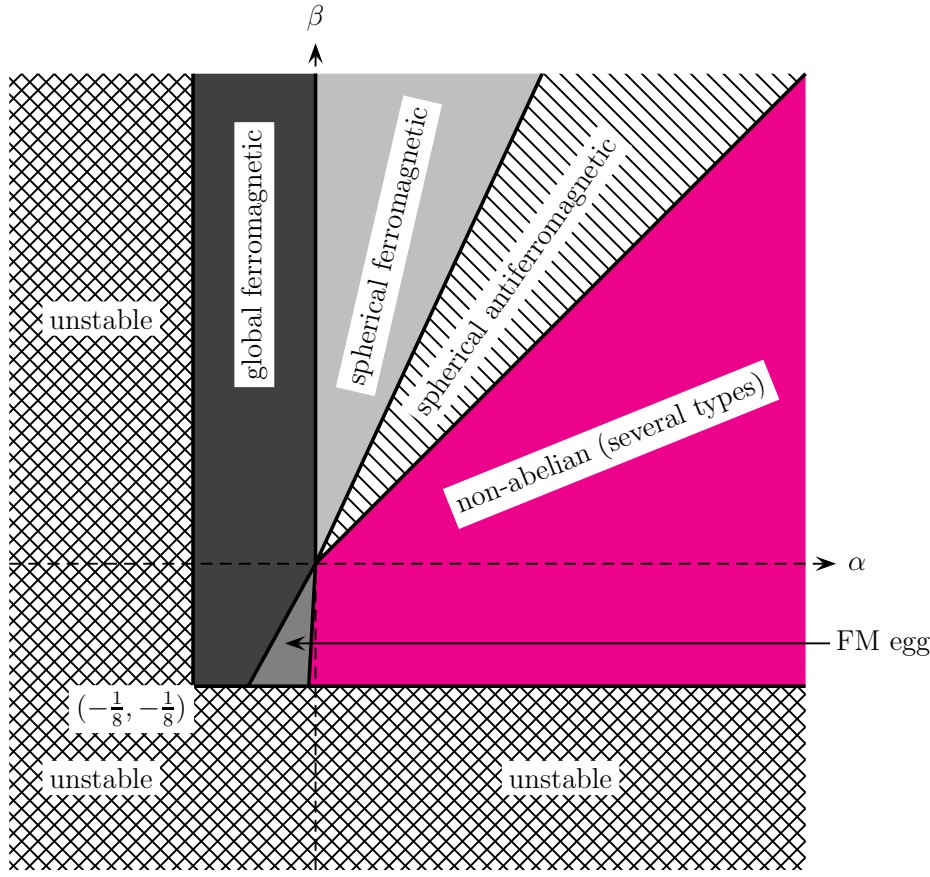

Figure 8: Orientation phase diagram of 3D quasi-instanton matter, regardless of lattice structure.

## 4.8  Summary

To summarize this whole section, the diagram of all IR-different orientation patterns for all allowed values of $\alpha, \beta > -\frac{1}{8}$ is shown on fig. 8.

# 5 The phase diagram of the non-Abelian crystals

We then continue to explore the phase diagram of the instanton crystals with the generalized interaction of eq. (9) depending on the parameters $\alpha$ and $\beta$. We have a more detailed look of the phase diagram in the non-Abelian phase, i.e., the magenta area of fig. 8. This phase is interesting because (unlike in the other phases) simulations suggest that various different crystal structures appear in this phase, giving rise to a highly nontrivial phase diagram. Moreover, these crystals are "regular" so that they leave some of the (discrete) translation symmetries intact, unlike the structures found in the spherical (anti-)ferromagnetic and mixed ferromagnetic phases. Therefore they admit a precise mathematical description, which makes further analysis possible.

We proceed as follows. Based on the simulation results, we identify the precise structure (including spatial lattice and orientations) of all crystals appearing in the non-Abelian phase. This allows us to generate numerically very large perfect crystal domains, having $\mathcal{O}\left(10^7\right)$ instantons. Then we can compute the energies of all these crystals to a high precision, and compare them in order to draw the phase diagram.

## 5.1 Non-Abelian crystal structures

By carrying out simulations in Sec. 3, we have identified several types of crystals as the ground states of the system. Up to changes in aspect ratios, they can be categorized into four different classes (excluding the hexagonal layers of fig. 4 (bottom left) which is found at large $\alpha$ and $\beta$; we restrict here to $\alpha$ and $\beta$ of at most $\mathcal{O}(1)$). The unit cells for these classes are shown in fig. 9, and the precise analytic definitions are the following:

**(a) Tetragonal/cubic (fcc) lattices with "standard" orientation pattern.** By this we mean fcc with the natural pattern made of four orientations $(1, i, j, k)$, and crystals which are obtained from this by rescaling the coordinate system in one spatial direction, so that the configuration depends on one aspect ratio. The crystals in fig. 2, fig. 3 (left), and fig. 4 (top right) fall into this class.

The explicit definition is as follows. We take a unit cell defined by the following three vectors

$$\vec{a}_1 = \vec{e}_1, \qquad \vec{a}_2 = \vec{e}_2, \qquad \vec{a}_3 = c\vec{e}_3, \tag{55}$$

where $\vec{e}_j$ are the unit vectors for the spatial coordinate system and $c$ is the aspect ratio. We then place the following four instantons

$$\vec{X}_1 = 0, \qquad\qquad y_1 = 1; \qquad \vec{X}_2 = \frac{1}{2}\vec{a}_1 + \frac{1}{2}\vec{a}_2, \qquad y_2 = k; \tag{56}$$

$$\vec{X}_3 = \frac{1}{2}\vec{a}_1 + \frac{1}{2}\vec{a}_3, \qquad y_3 = j; \qquad \vec{X}_4 = \frac{1}{2}\vec{a}_2 + \frac{1}{2}\vec{a}_3, \qquad y_4 = i. \tag{57}$$

The whole lattice is then obtained by carrying out shifts with linear combinations of $\vec{a}_j$ with integer coefficients. In fig. 9 (a) we show the defining instantons and all other instantons which touch the unit cell. In our conventions, fcc is obtained for $c = 1$ (which is the case shown in the figure) and body-centered-cubic (bcc) for $c = 1/\sqrt{2}$. In simulations, we have observed phases both with a wide range of values of $c$, including $c < 1$ (often close to the bcc value), $c = 1$ exactly, and $c > 1$. These crystals are denoted with blue color in the plots below.

**(b) Tetragonal/cubic lattices with "alternative" orientation pattern.** This system is otherwise the same as **(a)**, i.e. the spatial structure is obtained from fcc by scaling and the orien-

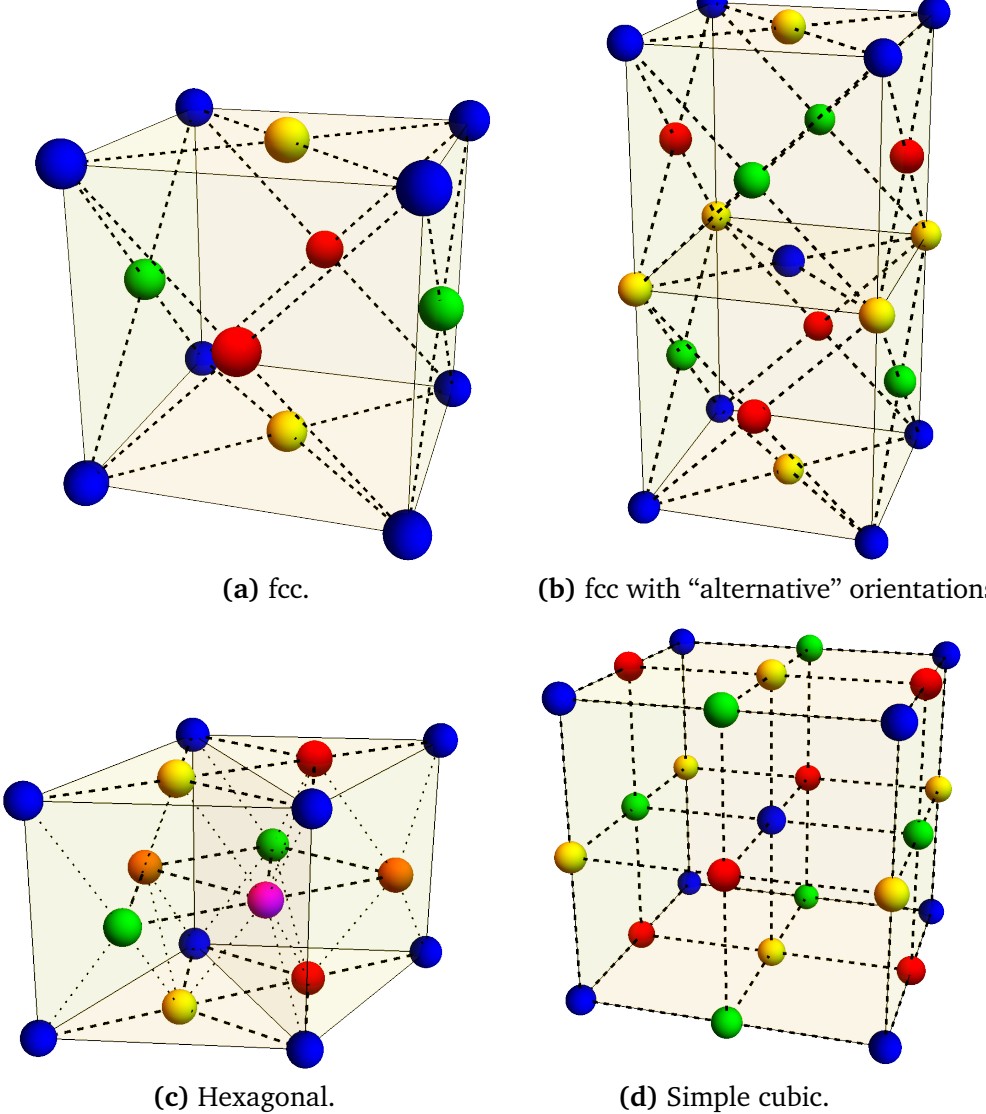

**(a)** fcc.  **(b)** fcc with "alternative" orientations.

**(c)** Hexagonal.  **(d)** Simple cubic.

Figure 9: The unit cells of the non-Abelian crystals. The crystal structures are the following: **(a)** Tetragonal/cubic (fcc) lattices with "standard" orientation pattern. **(b)** Tetragonal/cubic (fcc) lattices with "alternative" orientation pattern. **(c)** Hexagonal lattices (including hcp). **(c)** Simple tetragonal/cubic lattices. We show the results for the standard aspect ratio, i.e., $c = 1$ for the crystals **(a)**, **(b)** and **(d)** so that the lattices are exactly cubic, and $c = 2\sqrt{2}/3$ for the hexagonal case so that the lattice is hcp, but all crystals also appear with other values of the aspect ratio in the final phase diagrams. For the cubic lattices **(a)**, **(b)** and **(d)** blue, green, red, and yellow colors stand for the orientations $1$, $i$, $j$, and $k$, respectively. For the six orientations in the hexagonal case **(c)**, see text.

tations are $(1, i, j, k)$, but there is an additional shift in the orientations in alternating layers of the unit cells in $z$ direction. The crystal in fig. 4 (bottom right) falls into this class.

Precise definition of the lattice structure is as follows. The units cell now corresponds to two

unit cells of the fcc lattice,

$$\vec{a}_1 = \vec{e}_1 , \qquad \vec{a}_2 = \vec{e}_2 , \qquad \vec{a}_3 = 2c\vec{e}_3 . \tag{58}$$

The cell is then filled with the following eight instantons (see fig. 9 (b))[3]

$$\vec{X}_1 = 0 , \qquad y_1 = 1 ; \qquad \vec{X}_2 = \frac{1}{2}\vec{a}_1 + \frac{1}{2}\vec{a}_2 , \qquad y_2 = k ; \tag{59}$$

$$\vec{X}_3 = \frac{1}{2}\vec{a}_1 + \frac{1}{4}\vec{a}_3 , \qquad y_3 = j ; \qquad \vec{X}_4 = \frac{1}{2}\vec{a}_2 + \frac{1}{4}\vec{a}_3 , \qquad y_4 = i ; \tag{60}$$

$$\vec{X}_5 = \frac{1}{2}\vec{a}_3 , \qquad y_5 = k ; \qquad \vec{X}_6 = \frac{1}{2}\vec{a}_1 + \frac{1}{2}\vec{a}_2 + \frac{1}{2}\vec{a}_3 , \qquad y_6 = 1 ; \tag{61}$$

$$\vec{X}_7 = \frac{1}{2}\vec{a}_1 + \frac{3}{4}\vec{a}_3 , \qquad y_7 = i ; \qquad \vec{X}_8 = \frac{1}{2}\vec{a}_2 + \frac{3}{4}\vec{a}_3 , \qquad y_8 = j . \tag{62}$$

The whole lattice is again obtained by shifting with linear combinations of $\vec{a}_j$ with integer coefficients. We have found this lattice in simulations for $\beta \approx 0$ and with aspect ratio $c < 1$, being typically close to the bcc value of $1/\sqrt{2}$. These crystals are denoted with red color in the plots below.

**(c) Hexagonal lattices.** We have obtained as end points of the simulation lattices that are related to hexagonal-close-packed (hcp) lattice by a coordinate rescaling perpendicular to the triangular planes. The orientation pattern includes six different orientations which form two equilateral triangles. The crystal in fig. 4 (top left) falls in to this class.

The precise definition can be taken to be the following. The unit cell is defined in terms of

$$\vec{a}_1 = \vec{e}_1 , \qquad \vec{a}_2 = \frac{1}{2}\vec{e}_1 + \frac{\sqrt{3}}{2}\vec{e}_2 , \qquad \vec{a}_3 = c\vec{e}_3 . \tag{63}$$

Notice that for hexagonal systems the unit cell is not rectangular. Moreover our definitions do not follow the standard in the literature; this is because due to the orientation structure it is convenient to define a unit cell which is larger in the $\vec{e}_1$ and $\vec{e}_2$ directions than what is typically used for hexagonal lattices. The hcp lattice has $c = 2\sqrt{2}/3 \approx 0.9428$ in our conventions.

The orientations and locations of the defining six instantons are given by (see fig. 9 (c))

$$\vec{X}_1 = 0 , \qquad y_1 = 1 ; \qquad \vec{X}_2 = \frac{1}{3}\vec{a}_1 + \frac{1}{3}\vec{a}_2 , \qquad y_2 = \frac{1}{2} + \frac{\sqrt{3}}{2}k ; \tag{64}$$

$$\vec{X}_3 = \frac{2}{3}\vec{a}_1 + \frac{2}{3}\vec{a}_2 , \quad y_3 = \frac{1}{2} - \frac{\sqrt{3}}{2}k ; \qquad \vec{X}_4 = \frac{1}{3}\vec{a}_1 + \frac{1}{2}\vec{a}_3 , \qquad y_4 = j ; \tag{65}$$

$$\vec{X}_5 = \frac{2}{3}\vec{a}_2 + \frac{1}{2}\vec{a}_3 , \quad y_5 = \frac{\sqrt{3}}{2}i - \frac{1}{2}j ; \qquad \vec{X}_6 = \frac{2}{3}\vec{a}_1 + \frac{1}{3}\vec{a}_2 + \frac{1}{2}\vec{a}_3 , \quad y_6 = \frac{\sqrt{3}}{2}i + \frac{1}{2}j . \tag{66}$$

In simulations we have seen a variant of the lattice with the aspect ratio around $c \approx 0.5$. This kind of hexagonal lattice is found in nature in the tungsten carbide compound (which has $c \approx 0.57$). The comparison of energies done in this section reveals another variant, having $c$ close to the hcp value, is dominant in a narrow region on the $(\alpha, \beta)$-plane. Hexagonal crystals are denoted with green color in the plots below.

**(d) Simple cubic/tetragonal lattices.** Finally we have also seen non-Abelian simple cubic lattices in the region $\beta < 0$. The crystal in fig. 3 (right) falls in to this class. The precise definition for these lattices can be taken to be the following. The unit cell is defined in terms of

$$\vec{a}_1 = 2\vec{e}_1 , \qquad \vec{a}_2 = 2\vec{e}_2 , \qquad \vec{a}_3 = 2c\vec{e}_3 . \tag{67}$$

---

[3]It would be possible to define the crystal by using a unit cell with four instantons only but the cell would need to be non-rectangular.

Here we chose the units such that the shortest distances between the instantons in the directions $\vec{e}_1$ and $\vec{e}_2$ will be equal to one, but a factor of two was added due to the orientation structure, which will repeat in cycles of two for each spatial directions.

The orientations and locations of the defining eight instantons are given by (see fig. 9 (d))

$$\vec{X}_1 = 0 , \qquad y_1 = 1 ; \qquad \vec{X}_2 = \frac{1}{2}\vec{a}_3 , \qquad y_2 = k ; \qquad (68)$$

$$\vec{X}_3 = \frac{1}{2}\vec{a}_2 , \qquad y_3 = j ; \qquad \vec{X}_4 = \frac{1}{2}\vec{a}_1 , \qquad y_4 = i ; \qquad (69)$$

$$\vec{X}_5 = \frac{1}{2}\vec{a}_1 + \frac{1}{2}\vec{a}_2 , \qquad y_5 = k ; \qquad \vec{X}_6 = \frac{1}{2}\vec{a}_1 + \frac{1}{2}\vec{a}_3 , \qquad y_6 = j ; \qquad (70)$$

$$\vec{X}_7 = \frac{1}{2}\vec{a}_2 + \frac{1}{2}\vec{a}_3 , \qquad y_7 = i ; \qquad \vec{X}_8 = \frac{1}{2}\vec{a}_1 + \frac{1}{2}\vec{a}_2 + \frac{1}{2}\vec{a}_3 , \qquad y_8 = 1 . \qquad (71)$$

In simulations, we have only found lattices with $c = 1$ exactly. However, based on the analysis of the minimum energies of the various lattices, we expect that cubic lattices with $c \neq 1$ (both lattices with $c > 1$ and with $c < 1$) have the lowest energy near the critical value $\beta = -1/8$. All the simple cubic or tetragonal lattices are marked with yellow in the phase diagrams.

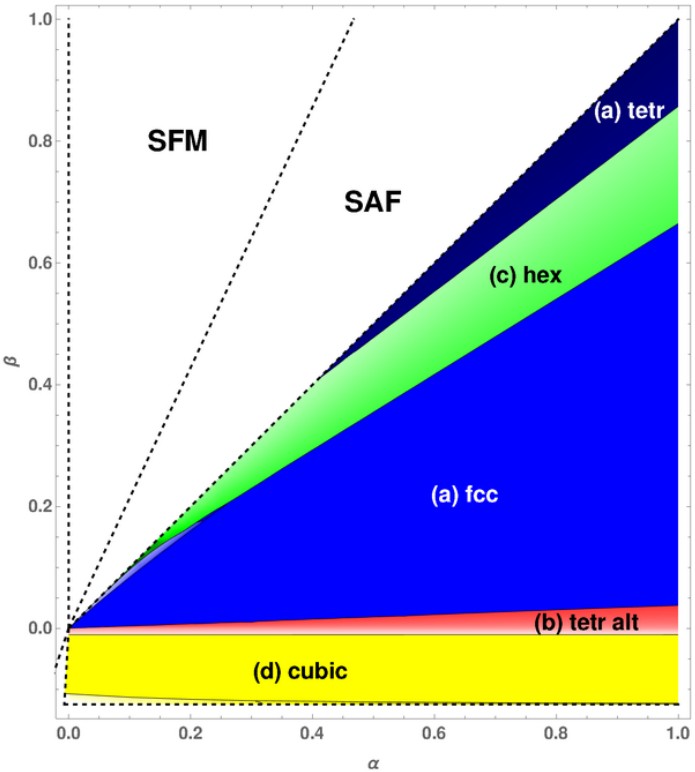

Figure 10: The phase diagram for the non-Abelian crystals. The blue, red, green, and yellow colors indicate crystals **(a)**, **(b)**, **(c)**, and **(d)** of fig. 9, respectively, as also shown in the labels. Changes in the brightness of the colors indicates changes in aspect ratios.

## 5.2 The phase diagrams

In order to carry out the detailed analysis of the non-Abelian phase, we computed numerically the energies of large lattices (with $\mathcal{O}\left(10^7\right)$ instantons) for the various crystal structures. See Appendix B for details of the computation. We included all the crystals of fig. 9 and scanned over all reasonable values of aspect ratios for each of them.

First, before going to the non-Abelian structure, we verify that the lowest energy configuration for the unoriented interaction of (7) is indeed fcc. The fcc, hcp, bcc, and simple cubic lattices are all local minima, with the energies satisfying $E(\text{fcc}) < E(\text{hcp}) < E(\text{bcc}) \ll E(\text{simple cubic})$. We also checked that the dominant crystal for the oriented, standard instanton interaction of (5) is the tetragonal (fcc-related) crystal having a large aspect ratio, which is shown in fig. 2. For the aspect ratio we find $c = 2.467$, a number slightly larger than that estimated from the simulations, $c \approx 2.35$.

We then draw the phase diagram of the non-Abelian lattices on the $(\alpha, \beta)$-plane. The result for the phase diagram is shown in fig. 10. In the figure the blue, red, green, and yellow colors indicate crystals **(a)**, **(b)**, **(c)**, and **(d)** of fig. 9, respectively. The brightness of the color is given by the aspect ratio, with brighter (darker) shades corresponding to lower (higher) values. The largest solid blue domain is fcc, i.e., aspect ratio is $c = 1$, and the large yellow domain is simple cubic. In all the other domains, the aspect ratio varies with $\alpha$ and $\beta$.

We continue by analyzing the phase diagram close to the critical lines already observed in the simulations in Sec. 3, starting from the results near the diagonal, $0 < \alpha = \beta < 1$.

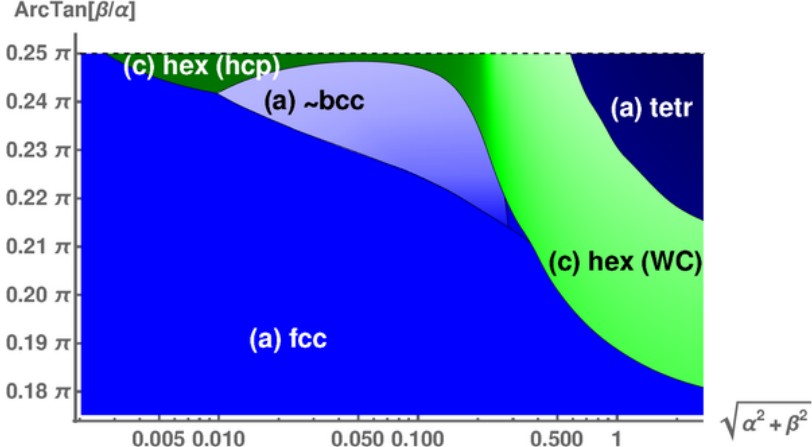

Figure 11: The details of the phase diagram near $\alpha = \beta$. We use angular coordinates and show the horizontal axis in logarithmic scale. Notation as in fig. 10.

We show the phase structure in this region in fig. 11, where we switched to angular coordinates and show the "radial" coordinate in log scale. In this area only the crystals **(a)** (fcc) and **(c)** (hexagonal) of fig. 9 appear, but the aspect ratios vary. In the light-blue domain the aspect ratio is close to the bcc value except for very close to the transition line to fcc and the tail at largest values of $\alpha$ within this domain. In the hexagonal green domain, the aspect ratio is close to the value of tungsten carbide (denoted by WC in the plot) at large $\alpha$ and $\beta$. As $\alpha$ decreases there is however a crossover to region where the aspect ratio is close to the hcp value of $2\sqrt{2}/3 \approx 0.94$. The crossover takes place at $\alpha \approx 0.15$. In the remaining dark blue tetragonal domain, the aspect ratio is large and increases with $\beta$, reaching the value of $c \approx 2.467$ at the physical point $\alpha = 1 = \beta$. Therefore in this region the structure is most naturally viewed as

aligned planes of anti-ferromagnetic square lattices with alternating orientations, as we also saw in simulations in Sec. 3.

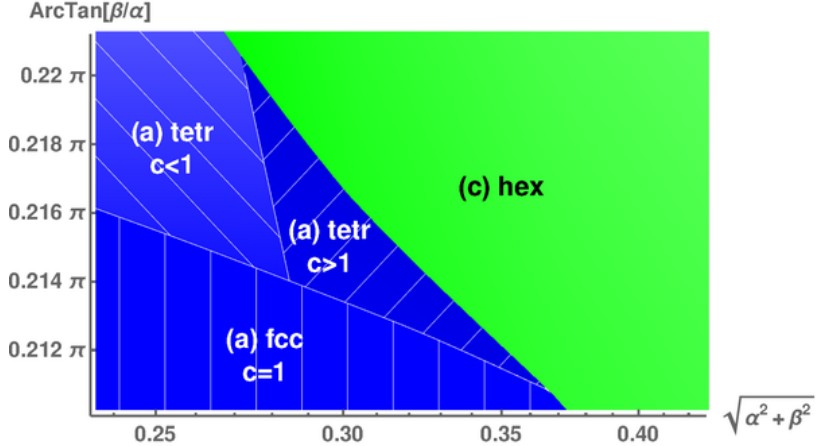

Figure 12: Zoom of the phase diagram in the critical region of fig. 11.

There are additional details in the area where the (near) bcc, fcc and hexagonal phases meet, which are poorly visible in fig. 11. To resolve the details, we show a zoom into this region in fig. 12. The blue regions in this plot are all versions of the same crystal, **(a)** in fig. 9, but having slightly different aspect ratios close to the fcc value $c = 1$. Since the aspect ratios are close, they are shown with similar colors. To make the various phases better visible we added stripes in the plot. All phase boundaries are first order transitions, but the transitions between the near fcc phases are weak. In particular the triple point is critical: transitions along generic lines intersecting the point are of second order.

Results for the energy differences and aspects ratios along the line $\alpha = \beta$ are shown in fig. 13. The plots on the top row show the energy difference $\Delta E$ between various phases and the fcc configuration (with aspect ratio $c = 1$) as a function of $\alpha = \beta$, in linear scale in the left plot and in log-log scale in the right plot. The "units" of $\Delta E$ depend on the system size (see Appendix B). The bottom row plot show the aspect ratio as a function of $\alpha$ in the various phases.

Along the line $\alpha = \beta$ there are two first order phase transitions, as we can also see from fig. 11. In the unoriented limit of very small $\alpha$, the fcc phase (with aspect ratio $c = 1$ exactly) is the ground state. At $\alpha \approx 0.0019$ we find a first order phase transition to a hexagonal phase (dot-dashed green curves). This phase transition is so close to the origin that it is not visible in the top left plot but it is identified as the point where the dot-dashed green curve hits zero on the top right log-log plot. At $\alpha \approx 0.41$ there is another transition, to the tetragonal lattice with large aspect ratio (dark blue dotted curves). This is the ground state up to $\alpha \sim 2$, including the physically interesting point $\alpha = 1$. It is surprise that we find hexagonal ground state at $\alpha \sim 0.1$, since the simulations of Sec. 3 would converge to a bcc structure in this region. Notice however that the energy difference between the hexagonal structure and the bcc (light blue dashed curves) is extremely slim. Apparently our simulations are not sensitive enough to discern this difference.

Notice also that the crossover in the hexagonal phase from approximate hcp lattice to tungsten carbide lattice at $\alpha \approx 0.15$ is clearly seen in the bottom plot: the aspect ratio in the hexagonal phase varies rapidly, starting from $c \sim 0.9$ at low $\alpha$ and ending at $c \sim 0.5$ around $\alpha = 0.4$. At the same time the energy in the top left plot shows a smooth kink around $\alpha \approx 0.15$ for the hexagonal phase. The value of the aspect ratio for the tungsten carbide compound appearing

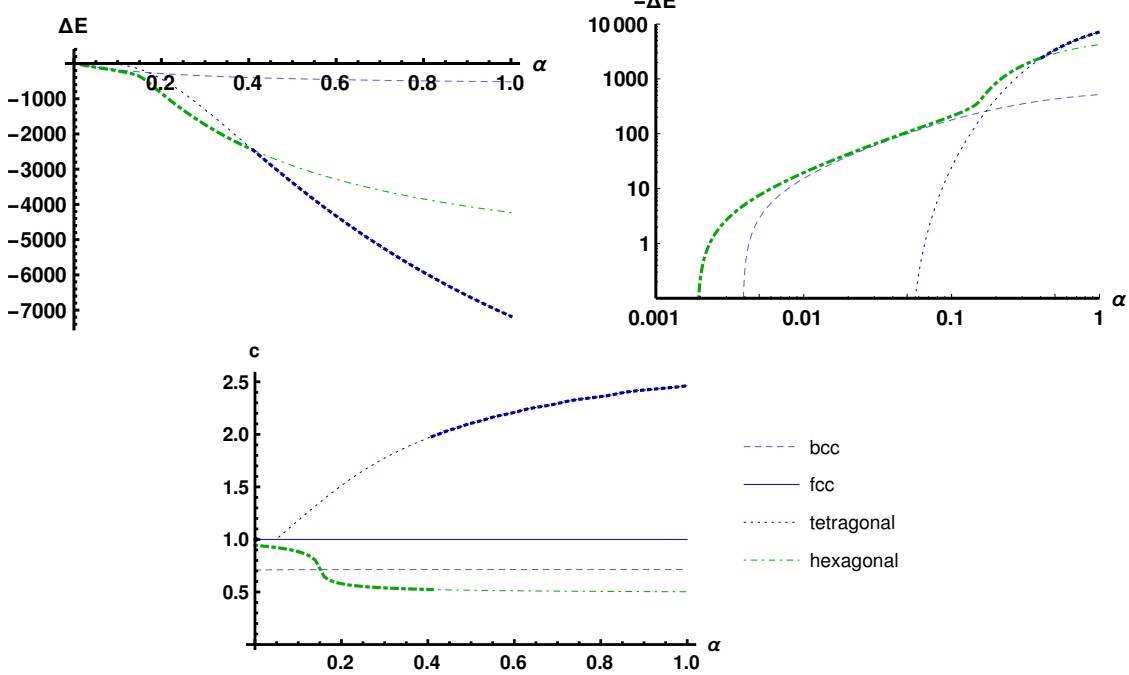

Figure 13: The non-Abelian crystals for $0 < \alpha = \beta < 1$. Top row: the energy differ-
ence of different phases with respect to the fcc configuration as a function of $\alpha$. Bot-
tom row: the aspect ratios as a function of $\alpha$. Blue colors indicate cubic/tetragonal
lattices, fig. 9 **(a)**, and green colors are hexagonal, fig. 9 **(c)**. In more detail, the
solid blue, dashed light blue, and dotted dark blue curves are the results for the
fcc, (near) bcc, and tetragonal large-$c$ lattices, respectively, whereas the dot-dashed
green curves show the result for the hexagonal lattices. Dominant phases are shown
as thick curves and subdominant as thin curves.

in nature, $c \approx 0.57$, is crossed at $\alpha \approx 0.21$.

Recall that in the fcc configuration (solid blue curve) $c = 1$ exactly due to the enhanced
symmetry, but e.g. for the bcc configuration (dashed blue curves in fig. 13 and the light blue
domain in fig. 11) the aspect ratio equals only approximately the bcc value $1/\sqrt{2}$, and depends
mildly on $\alpha$ and $\beta$. We have verified numerically, however, that $c \to 1/\sqrt{2}$ for the bcc config-
uration as $\alpha \to 0$ and $\beta \to 0$, which is natural as the effect of the orientations is suppressed
in this limit. Similarly in the hexagonal phase $c \to 2\sqrt{2}/3$ as $\alpha \to 0$ and $\beta \to 0$, which is the
value for hcp lattice in our conventions. Notice that neither the bcc configuration nor the hcp
configuration is the ground state at small values of $\alpha$ and $\beta$ but they still exists as subdominant
local minima, as seen from fig. 13.

We go on discussing the phase diagram in the vicinity of the line $\beta = 0$. As already seen
from fig. 10, in this region the tetragonal lattice with "alternative" orientation pattern marked
with red color (lattice **(b)** in fig. 9) dominates. We have zoomed into this region in fig. 14 by
using angular coordinates, in analogy to fig. 11.

We see that the tetragonal phase (red domain) is divided, at small $\alpha$, into two phases sep-
arated with a first order transition line ending at a critical point. The lower (upper) phase has
$c \approx 1/\sqrt{2}$ ($c \approx 1$) so the lattice is close to bcc (fcc), respectively. We have verified numerically

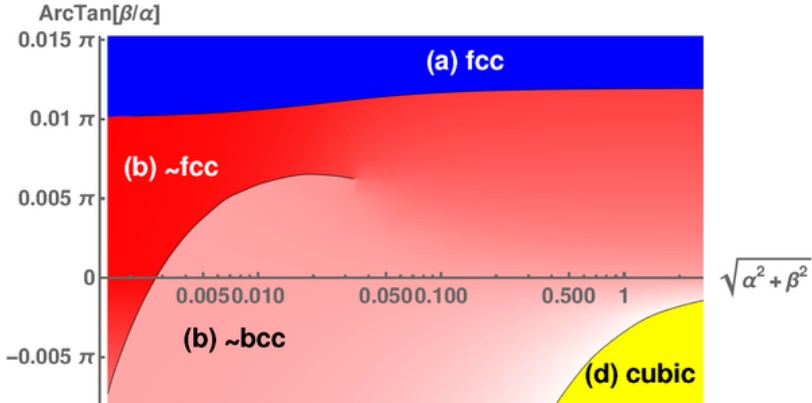

Figure 14: The phase diagram for the non-Abelian crystals near $\beta = 0$ in angular coordinates. Notice that the horizontal axis is in logarithmic scale. Notation as in fig. 10

that (apart from the section in the near fcc phase at very small $\alpha$) $c = 1/\sqrt{2}$ on the $\beta = 0$ line (the horizontal axis in the plot) so the lattice is exactly bcc on this line.

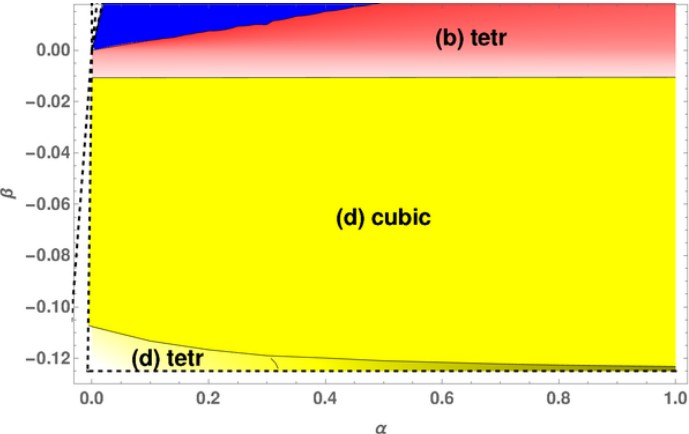

Figure 15: Zoom into the phase diagram for the non-Abelian crystals in the region of negative $\beta$.

Finally, we have a closer look at the region of negative $\beta$ in fig. 15. First, we notice that the transition line between the tetragonal phase (red domain) with alternative orientation pattern, and the simple cubic lattice (yellow domain) is almost precisely horizontal; we find that the transition takes place at $\beta \approx -0.011$. Second, there are two additional phases near the critical line $\beta = -1/8$. These phases have simple tetragonal structure, i.e., they have the orientation pattern of fig. 9 **(d)** but nontrivial aspect ratios. The phases were not found in simulations in Sec. 3, but the analysis of energies shows that they are favored with respect to the simple cubic in the vicinity of the critical line. The left phase (light yellow, smaller $\alpha$) has $c < 1$ and the right phase (dark yellow, larger $\alpha$) has $c > 1$. All the three yellow phases are separated by first order transitions, which become weak in the vicinity of the triple point at $\alpha \approx 0.3$. The diagram is therefore analogous to that of fig. 12, and the triple point is critical: transitions along lines passing though the point are, in general, of second order.

We also notice that, as expected, the orientation dependence of the final phase diagram

reflects the nature of the interaction terms (which was discussed in Sec. 2.2). First of all, as we already see from Fig. 8, negative $\alpha$ leads to ferromagnetic order, whereas for large positive $\alpha$ we encounter non-Abelian order. This is expected as positive (negative) $\alpha$ in the potential (9) favors different (the same) orientations. The somewhat more complicated effect of the $\beta$ term can also be discerned. For example the typical non-Abelian phases for $\beta > 0$ in Fig. 10 are the fcc or tetragonal phases **(a)**. From Fig. 9 we see that for nearest neighbor pairs, the twist $y_m^\dagger y_n$ is perpendicular to the link $\vec{N}_{mn}$ between the instantons (assuming the standard mapping from orientations to spatial directions). For the phases found at negative $\beta$, i.e., simple cubic or simple tetragonal phases **(d)**, the twist of any pair of nearest neighbors is parallel to the link between them.

# 6 Conclusions and open questions

The analysis of non-Abelian lattices and in particular those associated with large $N_c$ nuclear matter is in an infant stage and there are many questions to further investigate. Here we enlist some of them.

- The basic two body instanton interaction is with $\alpha = \beta = 1$. In this paper we have discovered a very rich structure of the phase diagram that followed from generalizing the interaction by varying the values of $\alpha$ and $\beta$. On top of the simulations described in this paper we have also performed several simulations in the region of $\alpha > 1$ or/and $\beta > 1$ and found interesting structure. It will be interesting to further study this region.

- The results of this paper are based on the generalized two instanton potential given in (9). A natural question is to what extent is this potential indeed general. In particular what is the form of the potential in the Skyrme and other nuclear matter models. A simple modification of the potential is achieved by using $\frac{1}{r^d}$ instead of $\frac{1}{r^2}$.

- Another obvious generalization is transforming the flavor symmetry $U(2) \to U(N_f)$ and in particular the eight-fold symmetry $N_f = 3$. In the opposite direction one would like to explore also the breaking of the isospin symmetry.

- The systems discussed in this paper are at zero temperature. It will be interesting to consider instead lattices of instantons (Torons) at finite temperature.

- We assumed here that the instantons were essentially point-like. Taking into account effects due to their finite size is a demanding task. Some nontrivial configurations are known however for the Skyrme models (see, e.g., [46–48]).

- The simulations performed in the paper have been made for instantons that were placed in a spherical cavity and a compensating "external" force was exerted on them to ensures that the sample has a constant density of instantons at large scales. We have made certain checks about the dependence of the final results on these two features and found very mild dependence mainly in the outer shells of instantons. This dependence on the structure of the cavity and the form of the external forces deserves further more elaborated study.

- In recent years there has been an effort to apply holography to the study of neutron starts and their mergers. It will be interesting to understand if and how taking crystal structures rather than the liquids change the picture of neutron stars. Moreover, our analysis

included the spherical fermionic and anti-fermionic structure. These phases may be relevant to the nuclear structure of neutron stars that obviously are spherically symmetric.

- Dressing the instantons with electric charges and differentiating between neutral and charged nucleons is a very important factor in real nuclei and nuclear matter. It is interesting to redo the analysis of the crystalline structure for chunks of holographic neutrons and protons.

- This research work dealt with uniform nuclear matter. It will be interesting to explore in what physical systems such uniformity can be realized. Needless to say that the study of non-uniform ensembles of instantons should also be of great interest.

## Acknowledgments

We would like to thank D. Melnikov for collaborations in previous research works on holographic nuclear matter. The work of M.J. and J.S. was supported in part by a center of excellence supported by the Israel Science Foundation (grant number 2289/18). The research of M.J. was also supported by an appointment to the JRG Program at the APCTP through the Science and Technology Promotion Fund and Lottery Fund of the Korean Government. In addition, the research of M.J. was supported by the Korean Local Governments — Gyeongsangbuk-do Province and Pohang City, and by the National Research Foundation of Korea (NRF) funded by the Korean government (MSIT) (grant number 2021R1A2C1010834).

## A   Potentials for a ball of homogeneous matter

In this Appendix we compute the potential of a spherical homogeneous configurations (i.e., ball of radius $R$) of constant density, corresponding to the continuum limits of the various phases encountered in the simulations. The setup is therefore similar to that analyzed in Sec. 4: The lattice geometry does not matter. The geometry-independent IR potential of a probe instanton scales like $V_{\text{IR}} \sim \rho R$, while the geometry-dependent contributions scale like $\Delta V \sim \rho a$, where $a$ is the lattice spacing. Therefore the geometry-dependent terms vanish in the continuum limit $a \rightarrow 0$. Moreover, we are taking an average over the orientations locally. More precisely, as explained in Sec. 4, it is enough to consider the average of the orientation dependent expression $Q_{12}$ of eq. (15).

These potentials are important because they are also used as an input for our numerical simulations: an external force is required for the (locally averaged) simulation results to have constant density. This amounts to the interactions between the instantons in the simulations and those outside the simulated ball (either exactly or up to some additional factors, depending on the phase). We discuss the interpretation of these potentials in more detail below.

### A.1   Potentials for the various orientation patterns

In general, the IR potential for a probe instanton located at point $\mathbf{x}_1$ (or rather average potential for the instantons in a small neighborhood of $\mathbf{x}_1$) inside a large ball of instanton matter of uniform density $\rho$ is

$$V_{\text{IR}}(\mathbf{x}_1) = \rho \int_{\text{ball}} d^3\mathbf{x}_2 \frac{\langle Q_{12} \rangle}{|\mathbf{x}_1 - \mathbf{x}_2|^2} . \tag{72}$$

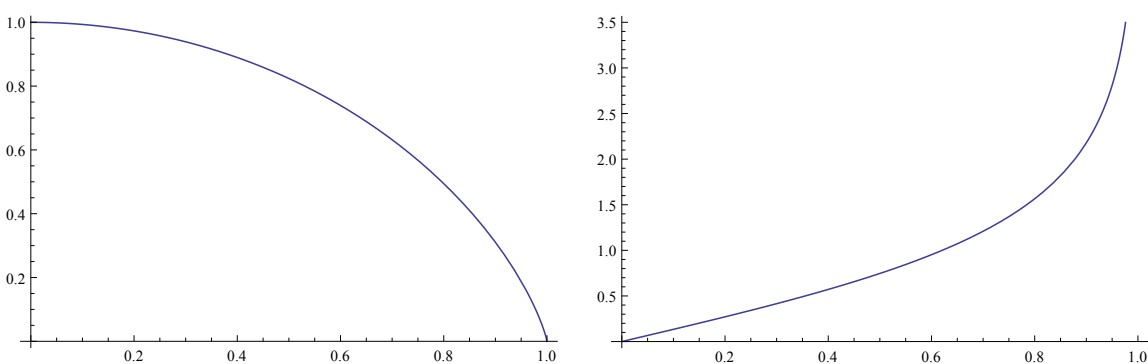

Figure 16: Left: The base potential $V_{\text{base}}(\mathbf{x})/2\pi\rho R$ as a function of $|\mathbf{x}|/R$. Right: $|\mathbf{F}_{\text{base}}(\mathbf{x})|/2\pi\rho R$ as a function of $|\mathbf{x}|/R$.

Further integrating over the location $\mathbf{x}_1$ of the probe within the ball of instanton matter would give us the net IR energy of all the instantons that we have analyzed in Sec. 4, specifically

$$E_{\text{IR}} \;=\; \frac{\rho}{2}\int_{\text{ball}} d^3\mathbf{x}_1\, V_{\text{IR}}(\mathbf{x}_1)\,. \tag{73}$$

But in this Appendix, we are interested in the potential (72) and the consequent bulk force

$$\mathbf{F}_{\text{bulk}}(\mathbf{x}) \;=\; -\nabla V_{\text{IR}}(\mathbf{x})\,, \tag{74}$$

which we would need to cancel by fiat in order to get a uniform instanton density in our numeric simulations.

Before we focus on specific orientation patterns, let's consider the baseline case of $\langle Q_{12}\rangle = 1$ and hence

$$V_{\text{base}}(\mathbf{x}_1) \;=\; \rho\int_{\substack{\text{ball of}\\\text{radius } R}} \frac{d^3\mathbf{x}_2}{|\mathbf{x}_1-\mathbf{x}_2|^2} \;=\; 2\pi\rho R\left(\frac{R^2-|\mathbf{x}_1|^2}{R|\mathbf{x}_1|}\operatorname{ar\,tanh}\frac{|\mathbf{x}_1|}{R}+1\right)\,. \tag{75}$$

This baseline potential is spherically symmetric, and it is easy to check that it is positive for all $r_1 < R$ and monotonically decreases with the radius $r_1 = |\mathbf{x}_1|$, see fig. 16 (left). Consequently, the bulk force (74) always points away from the center,

$$\mathbf{F}_{\text{base}}(\mathbf{x}) \;=\; 2\pi\rho\left(\left(\frac{R^2}{r^2}+1\right)\operatorname{ar\,tanh}\frac{r}{R}-\frac{R}{r}\right)\mathbf{n}\,. \tag{76}$$

The force is shown in 16 (right). Notice that it diverges logarithmically as $r \to R$. Given this baseline case, let's focus on the orientation patterns we have studied in section 4:

- *Non-Abelian patterns.* In this case we found that $\langle Q_{12}\rangle = \frac{1}{2} + \alpha + \beta$ in Sec. 4. Since this averaged coefficient of the $1/r^2$ force is constant, it follows that the IR potential is

$$V_{\text{IR}}^{\text{NA}}(\mathbf{x}_1) \;=\; \left(\frac{1}{2}+\alpha+\beta\right)\times V_{\text{base}}(\mathbf{x}_1)\,.$$

- *Global ferromagnetic pattern.* Again, we have a constant $\langle Q_{12}\rangle = \frac{1}{2} + 4\alpha$ (although a different constant from the NA patterns), hence the IR potential is

$$V_{\text{IR}}^{\text{FM}}(\mathbf{x}_1) \;=\; \left(\frac{1}{2}+4\alpha\right)\times V_{\text{base}}(\mathbf{x}_1)\,.$$

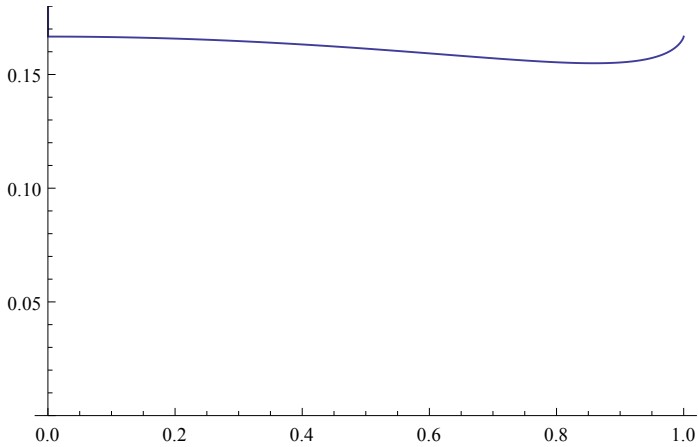

Figure 17: $V^{AF}_{\text{quadrupole}}(r)/2\pi\rho R$ as a function of $r/R$.

- *Global anti-ferromagnetic pattern.* For the global anti-ferromagnetic pattern,

$$\langle Q_{12} \rangle = \frac{1}{2} + 2\alpha + 2\beta(\mathbf{N}\cdot\mathbf{n}_{12})^2 . \tag{77}$$

This time, the averaged $\langle Q \rangle$ is not constant but direction dependent, so the IR potential for the probe instanton at $\mathbf{x}_1$ is not going to be spherically symmetric. We therefore start by averaging the expression (77) over the direction of the $\mathbf{x}_2$ but not over the $\mathbf{x}_1$. This gives us

$$\langle\langle Q_{12} \rangle\rangle = \frac{1}{2} + 2\alpha + \frac{2}{3}\beta + 2\beta\left(\frac{3}{2}\cos^2\theta_1 - \frac{1}{2}\right)\times\left[\frac{2}{3} - \frac{r_2^2\sin^2\theta_{12}}{r_1^2 + r_2^2 - 2r_1 r_2\cos\theta_{12}}\right], \tag{78}$$

where $\theta_{12}$ is the angle between the $\mathbf{x}_1$ and $\mathbf{x}_2$ directions while $\theta_1$ is the angle between the $\mathbf{x}_1$ and the direction $\mathbf{N}$ of the AF flip. The $\theta_1$ dependence of eq. (78) is a combination of a constant and a quadrupole, so the IR potential also has a spherically symmetric and a quadrupole terms,

$$V^{AF}_{\text{IR}}(\mathbf{x}_1) = \left(\frac{1}{2} + 2\alpha + \frac{2}{3}\beta\right)\times V_{\text{base}}(r_1) + 2\beta\left(\frac{3}{2}\cos^2\theta_1 - \frac{1}{2}\right)\times V^{AF}_{\text{quadrupole}}(r_1), \tag{79}$$

where

$$
\begin{aligned}
V^{AF}_{\text{quadrupole}}(r_1) &= \rho\int_{\text{big ball}} d^3\mathbf{x}_2\left[\frac{2/3}{|\mathbf{x}_1 - \mathbf{x}_2|^2} - \frac{r_2^2\sin^2\theta_{12}}{|\mathbf{x}_1 - \mathbf{x}_2|^4}\right] \\
&= 2\pi\rho\frac{3R^2 - r_1^2}{12r_1^3}\times\left(Rr_1 - (R^2 - r_1^2)\,\text{ar}\tanh\frac{r_1}{R}\right). \tag{80}
\end{aligned}
$$

The radial dependence of this quadrupole term is plotted on figure 17. As we can see, $V^{AF}_{\text{quadrupole}}$ is nearly constant within a $\pm5\%$ range, so it does not contribute much to the radial force. Instead, it cases a $1/r$ force in the longitudinal direction $\mathbf{n}_\theta$,

$$F^{AF}_\theta = 2\beta\frac{V^{AF}_{\text{quadrupole}}(r)}{r}\times 3\sin\theta\cos\theta \longrightarrow \frac{12\pi\beta\rho R}{r}\times\sin(2\theta) \quad\text{for } r\to 0. \tag{81}$$

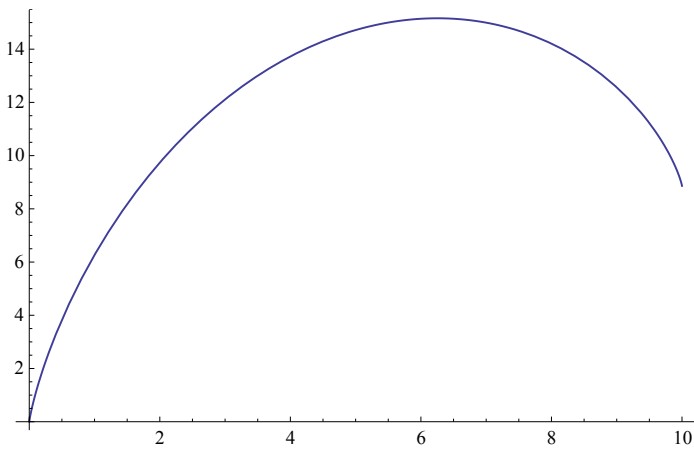

Figure 18: $\Delta_1 V(r_1)$ for $R = 10$.

- *Spherical anti-ferromagnetic pattern*. In this case we found that

$$\langle Q_{12} \rangle = \left( \frac{1}{2} + \alpha + \beta \right) + (\alpha - \beta) \times \cos \theta_{12} - 2\beta \times \frac{(r_1 - r_2 \cos \theta_{12})(r_2 - r_1 \cos \theta_{12})}{|\mathbf{x}_1 - \mathbf{x}_2|^2} . \quad (82)$$

Note that this $\langle Q_{12} \rangle$ depends on the angle $\theta_{12}$ between the $\mathbf{x}_1$ and $\mathbf{x}_2$ directions but is invariant under simultaneous rotation of both $\mathbf{x}_1$ and $\mathbf{x}_2$. Consequently, after integrating over the $\mathbf{x}_2$ we get a spherically symmetric IR potential for the instantons at $\mathbf{x}_1$. Specifically,

$$V_{\text{SAF}}^{\text{IR}}(r_1) = \left( \frac{1}{2} + \alpha + \beta \right) \times V_{\text{base}}(r_1) + (\alpha - \beta) \times \Delta_1 V(r_1) - 2\beta \times \Delta_2 V(r_1), \quad (83)$$

where

$$
\begin{aligned}
\Delta_1 V(r_1) &= \rho \int_{\text{big ball}} \frac{d^3 \mathbf{x}_2 \, \cos \theta_{12}}{|\mathbf{x}_1 - \mathbf{x}_2|^2} \\
&= \frac{4\pi\rho}{3} \times \left[ r_1 \log \frac{R^2 - r_1^2}{r_1^2} - \frac{R^2}{2r_1} + \frac{R(R^2 + 3r_1^2)}{2r_1^2} \operatorname{ar tanh} \frac{r}{R} \right],
\end{aligned}
$$

$$\quad (84)$$

which is drawn in fig. 18 is positive for all $r_1$, while the $\Delta_2 V$ happens to vanish,

$$\Delta_2 V(r_1) = \rho \int_{\text{big ball}} \frac{d^3 \mathbf{x}_2 \, (r_1 - r_2 \cos \theta_{12})(r_2 - r_1 \cos \theta_{12})}{|\mathbf{x}_1 - \mathbf{x}_2|^4} = 0 . \quad (85)$$

Note that for $\alpha = \beta$ the potential for the spherical AF setup is equal to that of the NA setup.

- *Spherical ferromagnetic pattern*. For this pattern, $\langle Q_{12} \rangle = \frac{1}{2} + 4\alpha \cos^2 \theta_{12}$. Note that $\langle Q \rangle$ depends only on the relative angle between the directions of $\mathbf{x}_1$ and $\mathbf{x}_2$, so integrating over the $\mathbf{x}_2$ gives us a spherically symmetric IR potential for an instanton at $\mathbf{x}_1$. Unfortunately, the explicit formula for this potential is rather messy,

$$V_{\text{SFM}}^{\text{IR}}(r_1) = \rho \int_{\text{big ball}} d^3 \mathbf{x}_2 \, \frac{\frac{1}{2} + 4\alpha \cos^2 \theta_{12}}{|\mathbf{x}_1 - \mathbf{x}_2|^2}$$

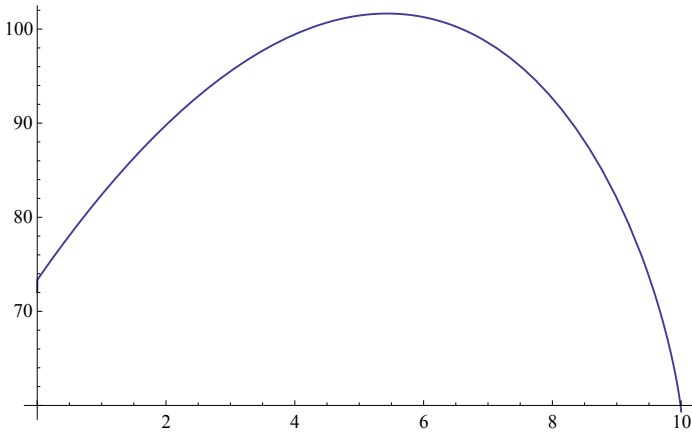

Figure 19: $V_{SFM}$ for $R = 10$ and $\alpha = 1$.

$$
= \pi\rho \begin{pmatrix} R & + \alpha\left(R - \frac{R^3}{r_1^2} + \pi^2 r_1\right) \\ + \frac{R^2 - r_1^2}{r_1^3}\left(r_1^2 + \alpha(R^2 + 5r_1^2)\right)\operatorname{ar tanh}\frac{r_1}{R} \\ + \alpha r_1\left(\operatorname{Li}_2\frac{r_1^2}{R^2} - 4\operatorname{Li}_2\frac{r_1}{R}\right) \end{pmatrix}.
$$
(86)

The potential is drawn in fig. 19 and the corresponding force is given by

$$
\begin{aligned}
\frac{1}{\pi\rho}F_{\text{SFM}} &= \alpha\left[\operatorname{Li}_2\left(\frac{r^2}{R^2}\right) - 4\operatorname{Li}_2\left(\frac{r}{R}\right) + \frac{3R^3}{r^3}\right. \\
&\quad \left. -\left(\frac{3R^4}{r^4} + \frac{4R^2}{r^2} + 9\right)\operatorname{ar tanh}\left(\frac{r}{R}\right) + \frac{5R}{r} + \pi^2\right] \\
&\quad -\left(\frac{R^2}{r^2} + 1\right)\operatorname{ar tanh}\left(\frac{r}{R}\right) + \frac{R}{r} \\
&= \pi^2\alpha - \frac{4(56\alpha + 5)r}{15R} + \mathcal{O}\left(r^2\right),
\end{aligned}
$$
(87)
(88)

which is drawn in fig. 20.

## A.2 Long-distance behavior of the potentials

Finally, let us comment on the long-distance cutoff dependence of these potentials. This is also linked to the fact that we are using the potentials to determine the external forces applied to the instantons in the simulations that we carry out. Actually, the natural choice for the external potential would be the potential of a homogeneous distribution (having a constant density which equals the average density of instantons in the simulation) outside the volume of simulation. However, due to the long-distance divergence of the two-body interaction, such a potential is not a priori well-defined, but depends on the long-distance cutoff.

For some of the potentials discussed above there is only one reasonable definition of the external potential, at least up to an irrelevant constant. These are the patters for which $\langle Q_{12}\rangle$ is constant, i.e., the non-Abelian and global ferromagnetic crystals. In this case the long-distance divergence for the homogeneous distribution is that of the plain $1/r^2$ interaction. Then the

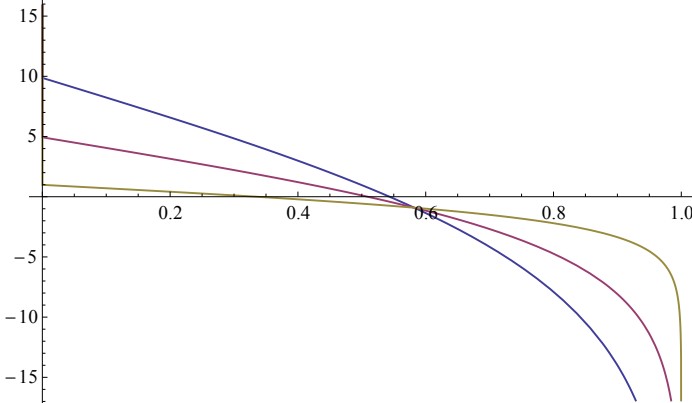

Figure 20: $F_{SFM}$ as a function of $\frac{r}{R}$ for $\alpha = 0.1$ (brown), $\alpha = 0.5$ (purple) and $\alpha = 1$ (blue)

result is independent of the precise definition of the cut-off, with the (reasonable) assumption that it does not introduce a "dipole" term (for example the cutoff is invariant as $\mathbf{x} \to -\mathbf{x}$). With such a cutoff and at constant density, the force due to homogeneous matter inside the ball of radius $R$ is exactly the opposite of the force due to matter outside the ball. This fact is reflected in the large $R$ expansion of the base potential,

$$V_{\text{base}}(r) = 4\pi\rho R + \mathcal{O}\left(\frac{1}{R}\right), \tag{89}$$

i.e., interpreting the value of $R$ as the long-distance cutoff. The only cutoff dependent piece is an irrelevant constant.

For the crystals having nontrivial long-range correlations between orientations, such as the spherical antiferromagnetic pattern, the above argument does not hold. Indeed, by expanding the potential at large $R$ we find that

$$V_{\text{SAF}}^{\text{IR}}(r) = 4\pi\rho\left(\frac{1}{2} + \alpha + \beta\right)R + \frac{4\pi\rho}{9}(\alpha - \beta)r\left(5 + 6\log\frac{R}{r}\right) + \mathcal{O}\left(\frac{1}{R}\right), \tag{90}$$

so that even with a simple spherical cutoff there is a nontrivial potential term depending on $R$. We also note that the force due to this potential is singular at the origin: it diverges as $\sim \log r$. Obviously such a divergent force field cannot be generated by any distribution of instantons outside the sphere, let alone a homogeneous distribution. Similar issues appear in the spherical ferromagnetic and ferromagnetic egg phases. Therefore the potentials which we use in the simulation, which sets the density of the instantons to (roughly) constant, are lacking a clear physical interpretation in these phases.

# B Numerical details

The numerical analysis of the configuration consists of two steps: First, we carry out simulations for various choices of the two-body potential, starting from a random initial condition and using an algorithm to seek for a minimum of total energy. We use a setup which roughly corresponds to adding masses and drag forces for the instantons, and simulating their dynamics as the system converges to a final state with low energy. We will explain this in more detail below. The likely

structure of the ground state can then be extracted from the final simulation results. The various crystals and the phase structure arising from this analysis are presented in Sec. 3. Second, for final states that are regular lattices, we can generate large configurations and compute their energies numerically directly without doing simulations. The energies can then be used to draw the phase diagram of the system as a function of the parameters of the interaction potential. The results from this analysis are used in Sec. 5 where we draw the final phase diagram of the non-Abelian crystals.

## B.1   Setup for simulations

We took $N$ instantons inside a three dimensional ball with radius $R$, starting from random initial positions $\vec{X}_k$ and random initial orientations $y_k$, and used an algorithm to minimize the total energy of the system. In order to reduce boundary effects, we also added an external force which sets the density of the instantons (averaged in a small neighborhood) to constant (see Appendix A). For the algorithm searching for the minimum of the energy, we implemented both a simple direct steepest descent method (first order formalism) and a second order formalism, which amounts to considering a dynamics for the instantons with a mass and a drag force and keeping track on their momentum. The second order formalism was seen to be clearly more efficient than the first order formalism.

We briefly sketch of the second order formalism which we use. For simplicity, we discuss mostly the relaxation of the positions (rather than orientations) of the instantons. Relaxing the system is an iterative procedure, which aims at minimizing the total energy

$$\mathcal{E}_{\text{tot}} = \sum_{n<m} \mathcal{E}^{2\,\text{body}}(n,m) + \sum_n \mathcal{E}_{\text{ext}}(n) , \tag{91}$$

where the two-body potential is given in (9) and the external potential is chosen such that the (locally averaged) density of the instantons is constant. The external potentials are given in Appendix A for all crystals we encountered. For the non-Abelian phase, where we did most of our simulations, it equals the interaction energy between the instanton $n$ and the unoriented continuum outside the ball. It is given (up to an irrelevant constant and normalization) by the base potential in (75),

$$\mathcal{E}_{\text{ext}}(n) = \frac{3NN_c}{2\lambda M} \frac{|X_n|}{R^3} \left(1 - \frac{R^2}{|X_n|^2}\right) \text{ar} \tanh \frac{|X_n|}{R} , \tag{92}$$

where the normalization was fixed by the requirement that density of the continuum matches with the average density of instantons inside the ball.

Let us then go to the details of the iterative algorithm which we used to minimize the energy. For each step in the procedure we compute a pseudo force as

$$\vec{F}_k = -\frac{\vec{\nabla}_{(k)} \mathcal{E}_{\text{tot}}}{\nabla^2_{(k)} \mathcal{E}_{\text{tot}}\big|_{\text{est}}} , \tag{93}$$

where the estimate in the denominator essentially averages over orientation dependence, i.e., we plug in

$$\nabla^2_{(k)} \mathcal{E}_{\text{tot}}\Big|_{\text{est}} = \sum_{n\neq k} \frac{6}{|X_n - X_k|^2} \mathcal{E}^{2\,\text{body}}(n,k) + \frac{\partial^2 \mathcal{E}_{\text{ext}}(k)}{\partial |X_k|^2} . \tag{94}$$

Here $\nabla_{(k)}$ acts on $\vec{X}_k$, and the first term is estimated by using the Laplacian for the unoriented potential ($\alpha = 0 = \beta$). We divide by this estimate because then the force (93) roughly measures the distance of the instanton $k$ from its equilibrium position when we are close to a local minimum. With this normalization it is easier to optimize the algorithm for the search of the minimum than when taking the force to be exactly the gradient of the energy.

A simple (first order) gradient descend would now simply iterate the positions simultaneously for all the instantons by using

$$\vec{X}_k^{\text{new}} = \vec{X}_k^{\text{old}} + \kappa \vec{F}_k\,, \tag{95}$$

where "old" and "new" refer to the positions of the instanton before and after the iterative update, respectively. A good guess for the coefficient $\kappa = \mathcal{O}(1)$ after normalizing the pseudo force as in (93). But as promised above, we implement a second order method. For this, we define a momentum $\vec{P}_k$ for each instanton which is initially set to zero. Then the iterative update is defined as

$$\begin{aligned} \vec{P}_k^{\text{new}} &= \kappa_P \vec{P}_k^{\text{old}} + \kappa_F \vec{F}_k\,, \tag{96} \\ \vec{X}_k^{\text{new}} &= \vec{X}_k^{\text{new}} + \vec{P}_k^{\text{new}}\,, \tag{97} \end{aligned}$$

where $\kappa_P$ and $\kappa_F$ are parameters which were tuned by trial and error. They correspond to adding masses and dissipation to the system. Notice that for $\kappa_P = 0$ one gets back the first order formalism, and one needs to have $0 \le \kappa_P < 1$; the limit of $\kappa_P \to 1$ from below is the limit of no dissipation. It turned out that good values for $\kappa_P$ are relatively close to one (meaning small dissipation); typically $\kappa_P = 0.99$ would lead to good convergence, i.e., a crystal with low total energy with a relatively low number of iteration steps. For $\kappa_F$, values close to the largest possible (where the system would still converge) were found to be optimal. In order to determine the optimal parameters (which slightly depend on both the system size and ground state crystal) we found it useful to monitor the total energy (91) while running the simulation.

The above algorithm will typically produce final state which is suggestive of crystal structure but has too many defects to definitely determine the crystal with lowest energy. In order to improve the quality, we add one more ingredient: a pseudo temperature. That is, we replace (93) by

$$\vec{F}_k = -\frac{\vec{\nabla}_{(k)} \mathcal{E}_{\text{tot}}}{\nabla_{(k)}^2 \mathcal{E}_{\text{tot}}\big|_{\text{est}}} + T \left(\nabla_{(k)}^2 \mathcal{E}_{\text{tot}}\big|_{\text{est}}\right)^{3/4} \vec{F}_{\text{rand}}\,, \tag{98}$$

where $T$ is the pseudo temperature and each component of the random force $\vec{F}_{\text{rand}}$ was taken from a uniform distribution between $-1$ and $1$.

A strategy which would produce crystal domains of good quality is the following. First let the system relax with zero temperature $T = 0$, repeating the update of (96) and (97), until the convergence stops so that the positions or orientations of the instantons no longer change. Then set $T$ to a value, found by trial and error, which just below the critical value which would completely melt the lattice structure. Then repeat the iteration procedure of (96) and (97) a large number of times, monitoring the total energy $\mathcal{E}_{\text{tot}}$, until it no longer decreases. As the last step, set the temperature again to zero and let the system to relax to a final state.

Finally let us comment on the relaxation in the orientation space. To implement this, we study the total energy in the vicinity of $y_k$ by defining

$$\mathcal{E}_{\text{tot}}(\vec{\epsilon}, k) = \mathcal{E}_{\text{tot}}\big|_{y_k \to y_k e^{i\vec{\epsilon}\cdot\vec{\tau}}}\,, \tag{99}$$

and then define a pseudo torque by

$$\vec{T}_k = -\frac{\vec{\nabla}\,\mathcal{E}_{\text{tot}}(\vec{\epsilon},k)\big|_{\vec{\epsilon}=0}}{\nabla^2\,\mathcal{E}_{\text{tot}}(\vec{\epsilon},k)\big|_{\text{est}}}\,, \tag{100}$$

where for the estimate we used simply

$$\nabla^2\,\mathcal{E}_{\text{tot}}(\vec{\epsilon},k)\big|_{\text{est}} = \sum_{n\neq k}\frac{N_c(\alpha+\beta)}{(1+2\alpha+2\beta)\lambda M}\frac{1}{|X_n-X_k|^2}\,. \tag{101}$$

The rest of the implementation is fully analogous to the position space approach. In particular we also implemented a second order algorithm for the orientation space even though this turned out to be much less important for convergence than the implementation in the position space. The complete algorithm then updates the orientations and (isospin) angular momenta of the instantons simultaneously with the positions and momenta of Eqs. (96) and (97).

We also remark that the external potential term in (91) depends on the final ground state crystal, which is not known before running the simulation. Therefore we first assumed the "standard" formula (92), and if the simulation would converge to a spherical ferromagnetic and antiferromagnetic crystal, we repeated the simulations taking the correct form for the external force and checked that the system converges to the same crystal.

### B.2  Computation of the energy differences

Here we discuss how we compute the precise phase diagram for the non-Abelian crystals. Once the precise structure of the crystals has been identified, as we have done in Sec. 5.1, it is in principle simple to compute numerically the energies of large samples of different crystals and compare them to draw the phase diagram. However, the size of the system is limited by computing resources, and because the two-body interaction leads to IR divergences for setups that are homogeneous at large scales, numerical noise due to the long distance cutoff in practice prevents a direct numerical comparison of the energies.

To overcome this issue we did the following. First, for all non-Abelian crystals which we have encountered, it is enough to compare the interaction energy of a single instanton with the rest of the sample. We choose this instanton to have unit orientation and to lie at the origin. We can then create large samples of various crystals including instantons up a radius $R$ with fixed density of instantons (which also include the instanton at the origin), and compute the energy

$$\mathcal{E}(R) = \sum_{n>1}\mathcal{E}^{2\,\text{body}}(n,1)\,, \tag{102}$$

where the index 1 refers to the instanton at the origin.

However even at the largest values of $R$ which we can handle, the function $\mathcal{E}(R)$ contains essentially random noise coming from how exactly the cutoff $r < R$ happens to select the instantons in the crystal structure. An efficient way to suppress this noise is the following. Instead of keeping $R$ fixed, we choose its value from a probability distribution (e.g. Gaussian) around some large value and with the deviation $\delta$ much larger than the spacing between the instantons. We then take an average over $R$ defined by this probability distribution. Choosing samples of $\mathcal{O}(10^7)$ instantons then turns out to be enough to remove essentially all noise from the results.

This approach boils down to computing the energy between the chosen instantons and the rest of the sample with a smooth large distance cutoff. That is, instead of (9) we use

$$\mathcal{E}^{2\,\text{body}}(m,n;\alpha,\beta;\Delta,\delta) \quad = \quad \frac{2N_c}{(1+2\alpha+2\beta)\lambda M}\frac{1}{|X_n-X_m|^2}$$

$$\times \quad \left[\frac{1}{2} + \alpha \operatorname{tr}^2\left(y_m^\dagger y_n\right) + \beta \operatorname{tr}^2\left(y_m^\dagger y_n(-i\vec{N}_{mn} \cdot \vec{\tau})\right)\right]$$

$$\times \quad \frac{1}{2}\left[1 - \tanh\left(\frac{|X_n - X_m| - \Delta}{\delta}\right)\right], \tag{103}$$

in (102) so that the interactions are cut off at distance $\Delta$ with deviation $\delta$. At $N_{\text{tot}} = 10^7$ instantons, generated in a ball with radius $R$, good choices turn out to be $\Delta = 0.7R$ and $\delta = 0.04R$.

One can show that the IR divergent term in the total energy is independent of the precise orientation structure and therefore exactly the same for all non-Abelian lattices at constant instanton density. The orientation dependent term is well convergent in the IR. We compute the energy differences between various non-Abelian lattices so that the divergent terms cancel, possibly up to terms depending on the details of the IR regulator. It is therefore enough to check that the results are independent of the parameters of the IR regulator to verify that the regulator-dependent terms are suppressed and the results are reliable. While it is presumably possible to do this analytically by using methods similar to those of Appendix A, we have only carried out a direct numerical check. Indeed we have verified numerically that the energy differences are insensitive to changes in $\Delta$ and $\delta$ to a very good precision when the parameters are varied in a reasonable range. Moreover we checked that applying an ellipsoidal cutoff instead of spherical, or changing the number of instantons does not change the results.

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
