# Peer review of "Many Phases of Generalized 3D Instanton Crystals"

_SciPost Physics, doi:SciPost Phys. 11, 018 (2021)_

## Round 2 · Referee Report · Anonymous (Referee 1) · 2021-3-15

Report

This paper discusses 3d instanton crystals for the purpose of studying the phase structures of nuclear matter. The paper performs a lattice simulation to handle a huge number of instantons to find out the lowest energy configurations. Although the reviewer doesn't make a full understanding of the details of the numerical computation, he finds that all the results are stated in a very well-organized and transparent manner. He thus recommends this paper to be published in SciPost. Before this, however, he asks the authors to add more comments or explanations about the following points, which he believes will help to improve the clarity of the paper.

(1) The paper takes into account only the two-body potential when analyzing the instanton crystals. This approximation seems valid when nucleons are well-separated. On the other hand, the two-body potential was derived originally for the short distance limit of two nucleons. Is there any common parameter regime where these approximations are consistent with each other?

(2) It would be preferable to discuss the physical meaning of the two parameters that appear in the generalized two-body potential.

(3) The paper discovers a phase such as the anti-ferromaganetic one, which exhibits a global spherical structure. It would be better to clarify if this is due to the boundary effect in the lattice simulation or not. The paper claims to observe a very strong tendency to converge to it. What are the implications the authors extract from these observations?

On top of these points, he reminds the authors that the layout of the pdf file is broken in his own PC environment(Mac+Preview), when he looks on the Figures from one to four.
  • validity: high
  • significance: high
  • originality: high
  • clarity: top
  • formatting: good
  • grammar: excellent

Author:  Matti Jarvinen  on 2021-04-27  [id 1386]

(in reply to Report 1 on 2021-03-15)

We thank the referee for careful reading of the manuscript and for the useful comments, and apologize for the slow response.

We will clarify the discussion of the numerical details in the second version of the manuscript.

Our response to the specific issues:

(1) The two body potential is motivated by the analysis in the Witten-Sakai-Sugimoto model. In this context there is indeed a parametrically large range where our expression for the potential is valid. This requires that the separation $d$ between the instantons satisfies $ 1/(M\sqrt{\lambda}) \ll d \ll 1/M$ where $\lambda$ is the coupling constant and $M$ is the scale of strong interactions. The lower limit arises from the instanton size and the upper limit from neglecting curvature corrections in the AdS geometry. In the strong coupling limit $\lambda \to \infty$, where the classical dual description is valid, these two scales are therefore well separated. This is discussed in detail in ref. [15] of the manuscript.

We notice that this point is not explained in the current version of the manuscript. We suggest to add an explanation in section 2.2 where the two-body potential is introduced.

(2) We agree that there should be more discussion of the interpretation of the parameters. Basic properties of the potential can be read from the two-body interaction term, eq. (8). The $\alpha$ coefficient multiplies a term where the dependence on orientation is independent of the spatial interactions. For positive (negative) $\alpha$ perpendicular (parallel) spins of nearby neighbors are preferred, and the effect increases with increasing $|\alpha|$. The interaction involving $\beta$ is a bit more complicated since there is nontrivial coupling between the orientations and directions in coordinate space. Picking an instanton with unit orientation $y=1$, positive (negative) $\beta$ means that orientation $y$ of a neighboring instanton which is perpendicular (parallel) to the spatial link between the two instantons is preferred (with the understanding that e.g. the orientation $y=i$ is parallel to the $x$ axis in coordinate space).

There are some scattered comments in sections 3 and 4 about the interactions but we plan to add a more detailed discussion in section 2.2. We will also comment how the final results for the phase diagram reflect the structure of the interaction term in section 5.

(3) The presence of the spherical phases is indeed a nontrivial question. We have carried out various checks (see below) some of which are mentioned in the manuscript. Based on them, we are confident that the spherical configurations are dominant over "regular" crystals in the regions indicated in figure 8. In particular, our analytic findings (which are independent of the boundary effects) agree to a great degree with the simulation results.

However it might happen that the true ground state in the continuum limit is some other, even more complicated structure (which we do not see in our simulations due to finite size/boundary effects) -- this we do not know. But even in with the largest volumes we could with relative ease simulate on a personal computer (about 40000 instantons), we did not see signs of any other structures. In addition we have checked that the spherical antiferromagnetic structure is insensitive to the shape of the cavity where we carry out the simulations. We have tried simulating i.e. in a cubic cavity and still found configurations which are clearly spherical near the center of the simulation result.

The strong tendency to form spherical structures for $\beta>\alpha>0$ seen in the simulations (the simulations converge to the final state more than 10 times faster than for $\alpha>\beta>0$) may mean that such configurations have a large energy gap with respect to regular crystals. This is in part confirmed by the IR analysis of the configurations.

We suggest to add more discussion of these points in section 3.4.

We also thank the referee for pointing out the issue with Preview. However, we did not observe issues in our own tests with this viewer, so we cannot fix the problem. (The 3D content in the figures is not available on Preview, and to our knowledge it only works with Adobe Reader.)

For the authors,
Matti Jarvinen

Anonymous on 2021-04-30  [id 1395]

(in reply to Matti Jarvinen on 2021-04-27 [id 1386])

The reviewer agrees the revised version to be published with no further revision.

---

## Round 2 · Referee Report · Anonymous (Referee 2) · 2021-4-27

Report

The authors consider the holographic description of nuclear matter as an ensemble of point-like instantons interacting with a pair interaction Eq. 1, that depends on the instanton relative distances and "flavor" orientations, but is essentially repulsive. The authors suggest to parametrize the flavor dependent and repulsive part using two positive parameters alpha and beta. Their numerical studies show rich lattice structures for varying alpha and beta. For alpha=beta=0 the canonical structure is fcc.

1/ Originally, Eq. 1 describes instantons interactions in 3-spatial dimensions plus 1-holographic dimension, leading to 4-dimensional crystal arrangements. However, the authors consider only 3-spatial dimensions. This important simplification should be explained.

2/ The long range sensitivity of the pair interaction in Eq. 1 noted by the authors, is naturally cutoff in nuclear matter by the the pion mass. Eq. 1 falls short of the long range pion attraction as the authors say in their introduction, and also is in the chiral limit. So clearly, without any external force the system will fly apart. To fix this, the authors implement a mean-field-like "attractive" force. The energetics of their crystalline structures depend on this mean-field, making certain structures energetically unfavorable in comparison to others. The authors should give a better understanding of these points.

Overall, the paper ideas and objectives are well stated. The results are interesting on their own for an ensemble of gauge quasi-particles pair interacting via Eq. 1 and worth publishing.

  • validity: -
  • significance: -
  • originality: -
  • clarity: -
  • formatting: -
  • grammar: -

Author:  Matti Jarvinen  on 2021-06-21  [id 1513]

(in reply to Report 2 on 2021-04-27)

We thank the referee for report and for the useful comments. Our response:

1/ This was not explained well in the first version of the manuscript and we thank the referee for pointing this out. Indeed also four dimensional crystals are possible. However at low density, one expects that the solitons are confined to a plane in the holographic direction so that the only relevant dimensions are the spatial ones. As the density increases, we expect a "popcorn" transition from three dimensional to four dimensional crystals, similar to the one to two dimensional transition considered, for example, in Ref. [15]. Studying the four dimensional crystals and the transition to them would be an interesting (but extremely challenging) topic for future research. There is a parametrically large region of densities where the three dimensional crystals can be embedded in the Witten-Sakai-Sugimoto setup. We have clarified the setup by adding comments on the 3D to 4D transition in the introduction on page 3 before Eq. (1) and in the beginning of Sec. 2.

2/ Indeed due to the long range sensitivity of the two-body interaction, we add an additional external force in our simulations. In the probably most interesting cases (i.e. the non-Abelian crystals) the external force is however well-defined: there is only a single choice which makes sense. This choice is the external force due to an unoriented continuum of instantons outside the cavity of the simulation. We stress that this is not an ad-hoc force but added because of the setup that we want to consider: in principle we want to consider an infinite sample, but due to limited resources, we can only simulate a relatively small bubble. The external force is then the mean field force due to all those instantons which were left out of the simulation due to this limitation. For the non-Abelian (and also for the global ferromagnetic) crystals this external force also leads to crystals having constant instanton density (approximated over a large enough volume), and we see no other reasonable choice for the force in this case.

For the oriented (spherical antiferromagnetic or ferromagnetic) phases, the situation is a bit more complicated. This is because the force which leads to constant instanton density is not the same as that obtained by averaging over an (anti)ferromagnetic continuum of instantons outside the cavity of simulations. Here we have chosen the external force such that the averaged instanton density is constant. This makes it possible to carry out the IR analysis of section 4. However we have also tried simulating with other choices of the eternal forces (these attempts are not documented in detail in the article for brevity), and did not see any significant changes in the results for the crystal structures locally. Globally, there were some changes, as the density of the instanton varies are one easily develops even empty regions in the middle of the cavity.

Notice that one can also do simulation without external force, even though this is presumably less well physically motivated. In this case one some of the solitons first cluster at the surface of the simulation volume, and then one obtains crystal formation in the middle. Even though we have not studied such configurations in detail, we expect that the crystal structures would be the same even in this case as with the simulations with the additional external force.

In summary, we do not expect that any of our main results are sensitive to the choices for the external force.

We have improved the discussion of the external force in the introduction (page 4, the first item (i) of the list and discussion before it), in the first paragraph of Sec. 3, and at the end of Sec 3.4. We now point out more clearly that the external force for the non-Abelian case appears to be essentially unique, and have added references to Appendix A.2, where a detailed discussion of the forces is given.

For the authors, Matti Jarvinen

---

## Round 3 · Author Response

We thank the referees for carefully reading the manuscript and for relevant comments that helped us to improve its quality.

The new version takes into account all comments of the invited referees, and fixed a few minor issues that we noticed ourselves. Please see the responses to the referee reports and the list of changes for details.

---

## Round 3 · List of Changes

The page and equation numbering in this list refers to the new (second) version of the manuscript.

  • We added a comment on the dimensionality of the crystals considered in the article in the last paragraph of page 3, before Eq. (1).
  • We revised the discussion of the external force in the third paragraph of page 4, and also modified the first item of the list in the fourth (last) paragraph on the same page, which contained an imprecise statement of the force.
  • We added a comment that detailed definitions will be given in Sec. 5 in the first item (oriented non-Abelian phase) of the second bulleted list on page 5.
  • We added the definition of bcc in the same item, near the end of page 5.
  • We added a comment on the nature and dimensionality of the instanton system in the first sentence of Sec. 2, end of page 6.
  • We added a discussion on how the two-body potential arises from the WSS model in the last paragraph of page 7. This includes the new Eq. (4).
  • We added a new paragraph at the end of Sec. 2.2, page 8, which discusses the interpretation of the parameters alpha and beta.
  • We added a comment on the external force and link to Appendix A.2, where it is discussed in detail, in the first paragraph of Sec. 3, on page 9.
  • We added a reminder that the details of the phase diagram will be discussed in Secs. 4 and 5 in the first paragraph of Sec. 3.3, near the end of page 12.
  • We added a clarifying comment on the rapid convergence in the spherical phases in the first paragraph of Sec. 3.4, on page 15.
  • We added a new paragraph discussing the choice of external force and on the various checks we have carried out in the spherically arranged phases at the end of Sec. 3.4, on page 16.
  • We removed an incomplete sentence in Sec. 5.1, page 29, right after Eq. (66).
  • We added a new paragraph which points out how the parameter dependence of the phase diagram reflects the changes in the two-body potential at the end of Sec. 5, on page 34.
  • We introduced new subsections titles (A.1 and A.2) in Appendix A so that we can refer to the discussion of the external force in Appendix A.2 more precisely.
  • We thoroughly revised Appendix B, adding several clarifying comments, cross links, and discussion on tricky points in the simulation.

---

## Editorial Decision

published